# Spectroscopic real-time monitoring of NO$_2$ for city scale modelling

P. Morten Hundt[1], Michael Müller[1], Markus Mangold[1,3], Béla Tuzson[1], Philipp Scheidegger[1], Herbert Looser[1,2], Christoph Hüglin[1] and Lukas Emmenegger[1]

[1]Laboratory for Air Pollution and Environmental Technology, Empa, 8600 Dübendorf, Switzerland
[2]Institute for Aerosol and Sensor Technology, FHNW, 5210 Windisch, Switzerland
[3]*Present address*: IRsweep AG, 8093 Zürich, Switzerland

*Correspondence to*: P. Morten Hundt (Morten.Hundt@empa.ch)

**Abstract.** Detailed knowledge about the urban NO$_2$ concentration field is a key element for obtaining accurate, individual exposure estimates. These are required for improving the understanding of the impact of ambient NO$_2$ on human health and for related air quality measures. We developed a compact and robust quantum cascade laser absorption spectrometer (QCLAS) and deployed it on a tram in the city of Zurich (Switzerland) to perform mobile real-time concentration measurements of NO$_2$ by direct absorption spectroscopy at 1600 cm$^{-1}$. Thorough analysis of the obtained NO$_2$ data, for instance by comparison with data from fixed air quality monitoring (AQM) sites, revealed the instrument to be highly accurate and valuable for collection of data that can be used in statistical models for the calculation of spatio-temporally resolved NO$_2$ concentration maps. The combination of fast mobile measurements with AQM data proved to be very suitable, but the statistical data analysis also showed that a single mobile instrument is not sufficient in the studied urban area, for mainly two reasons: (i) short residence close to sources with large short-term NO$_2$ variations and (ii) limited representativeness of the tram tracks for the entire urban environment.

## 1 Introduction

Numerous studies relate the exposure to nitrogen dioxide (NO$_2$) to adverse health effects (e.g. (Adam et al., 2014; Gehring et al., 2013; WHO, 2013)). Despite this threat to human health, limit values are regularly exceeded in European cities, mainly at locations directly impacted by traffic emissions (EEA, 2016). It is, therefore, highly relevant to provide spatially and temporally resolved NO$_2$ fields for the assessment of related health effects, for the guidance of efficient air quality measures and for urban air quality planning. True exposure of an individual is composed of the encountered pollutant concentration at a particular location and time and breathing rate. So far, the individual's exposure is mostly derived from Land Use Regression models representing seasonal or annual mean concentrations and the individual's home (and working) address (e.g. (Brauer et al., 2008; Cyrys et al., 2012)). Obviously, these values may be significantly biased depending on the mobility pattern of an individual as NO$_2$ concentrations in the urban environment are highly variable in space and time.

Spatially and temporally highly resolved pollution maps (< 20 m, < 1 hour) based on statistical modelling can enhance the accuracy of exposure estimates. However, such statistical models require accurate input data that represent the concentration levels at a set of different locations in an adequate temporal resolution. Until now, accurate (±1 ppb) and continuous NO$_2$ measurements are mainly performed at air quality monitoring (AQM) stations equipped with chemiluminescence detectors (CLD). Data from such AQM stations (e.g. seven locations in Zurich) do not provide sufficient spatial resolution for detailed pollution maps due to the high variability of influencing factors such as the traffic situation and the built environment.

One approach to overcome this challenge is the deployment of denser measurement networks. Mueller et al. (Mueller et al., 2015), for instance, used data from a dense network of passive diffusion samplers in Zurich (Switzerland) as input for statistical models. Such passive samplers, however, provide only average values over their exposure periods, which are days to weeks, and therefore lack the necessary temporal resolution. The replacement of passive samplers by low-cost

electrochemical sensors is being investigated but has not yet shown to be feasible for long-term deployment (Mead et al., 2013).

Mobile measurements are another option for increasing the spatial resolution (Hagemann et al., 2014; Hasenfratz et al., 2015; Kehl, 2007). They require highly sensitive – typical concentrations range from 5 to 200 pbb – and selective measurement devices with temporal resolutions of just a few seconds. With CLDs, $NO_2$ is determined indirectly as the

difference of consecutively measured NOx and NO, making this technique not fast enough for mobile measurements as their response time is longer than 1 minute. The indirect measurement approach of the CLD instruments can lead to biased $NO_2$ measurements when NO and $NO_2$ are temporally highly variable. Furthermore, the conversion of $NO_2$ to NO in a catalytic converter can lead to measurement errors as other species such as $NH_3$, $HNO_3$ and $HNO_2$ can be converted, too, and therefore lead to $NO_2$ signals that are biased high (NABEL, 2016; Steinbacher et al., 2007). Also electrochemical sensors are

currently not suited for such applications as their response time with about 60 seconds is too long. Laser spectroscopy, in contrast to CLD and electrochemical sensors, determines the concentration of $NO_2$ directly and with sampling rates of 1 Hz and higher. Previous studies have shown the applicability of laser diodes in the blue spectral region to measuring $NO_2$ by various cavity enhanced techniques (Courtillot et al., 2006; Fuchs et al., 2009; Kebabian et al., 2005; Osthoff et al., 2006). With higher absorption cross sections than in the visible the mid-infrared region is often chosen for high precision trace gas

monitoring of $NO_2$. Here, for instance photoacoustic spectroscopy (Pushkarsky et al., 2006), quartz-enhanced photoacoustic spectroscopy (Patimisco et al., 2014) and direct absorption spectroscopy (McManus et al., 2015) were applied.

Our method of choice is mid-infrared direct laser absorption spectroscopy at 1600 cm$^{-1}$ where the concentration of $NO_2$ is determined directly by measuring the absorption by ro-vibrational transitions of the $NO_2$ molecule. This method is highly selective and very sensitive with detection limits of only a few ppt (McManus et al., 2015). Tuzson et al. (Tuzson et al.,

2013), for example, deployed a two laser quantum cascade laser spectrometer (QCLAS) on the high-altitude air monitoring site Jungfraujoch (3580 m a.s.l., Switzerland) to measure background concentrations of NO and $NO_2$. Additionally, laser spectroscopy allows high sampling rates as shown e.g. by Jagerska et al. (Jagerska et al., 2015) for simultaneous NO and $NO_2$ measurement in engine exhaust gas with a sampling rate of 10 Hz. Furthermore, compared to other spectroscopic techniques, direct absorption spectroscopy relies on a relatively simple optical layout comprised of a light source, a long-

path cell and an infrared detector.

Therefore, our approach to perform accurate, mobile and direct measurements of $NO_2$ is to deploy a compact and robust quantum cascade laser (QCLAS) spectrometer on the roof of a tram that is operating as public transport service in the city of Zurich. The $NO_2$ concentration is measured by direct absorption spectroscopy at 1600 cm$^{-1}$. In this article, we present our newly developed laser spectrometer, analyze the performance of the instrument and quantify the accuracy of the $NO_2$

measurements. Moreover, $NO_2$ measurements from the single mobile instrument are investigated with respect to its use in highly resolved statistical models. Our study uses the extensive data set from air quality monitoring facilities in the city of

Zurich. The information about instantaneous $NO_2$ concentrations obtained with fixed air quality instruments provides the opportunity to validate the tram based measurements as well as the model predictions.

## 2 Methods

### 2.1 Quantum cascade laser spectrometer

The portable instrument was specifically built for this campaign. Figure 1 shows a photograph of the QCLAS instrument without cover and a schematic of the instrument and the optical layout. The whole instrument is built in a $40 \times 36 \times 15$ cm waterproof box and mounted on four metal springs which absorb vibrations from the moving tram. The optics are mounted on a carbon fiber breadboard which is supported by additional metal springs to further reduce vibrations. The instrument weights about 10 kg and is powered by a 12 V connection supplied by the tram. $NO_2$ concentrations are determined by

measuring the direct absorption signal of a single ro-vibrational transition in the mid-infrared. The main components of the instrument are a single mode (distributed feedback) DFB QCL (Alpes Lasers), a cylindrical multipass cell (Mangold et al., 2016) and a thermoelectrically cooled MCT detector (PVI-4TE-6, Vigo Systems). The laser is packaged in a high-heat-load (HHL) housing, where a Peltier element is used for temperature control. The QCL is operated at room temperature and emits 3 mW at 1600 cm$^{-1}$. It is driven in intermittent continuous-wave (iCW) mode (Fischer et al., 2014) with a pulse duration of

160 µs and a duty cycle of 50 %. In iCW mode the driving current of the QCL is dropped to zero between individual pulses leading to rapid heating and therefore frequency tuning during each pulse. The pulses are generated by discharging a capacitor over the QCL. A frequency tuning of >1 cm$^{-1}$ was achieved (see Figure 1(c)). By shaping the current ramp with different RC-elements, close to linear tuning could be achieved. See (Fischer et al., 2014) for details. In this driving mode the heat dissipation of the laser is lowered such that a fan is sufficient to cool the laser housing, and thus no additional

cooling liquid circulation is needed.

The output of the QCL is collimated inside the HHL housing by a f=1.873mm aspheric lens and focused into the multipass cell at half way of the first pass with a CaF$_2$ lens (f=150 mm). For frequency calibration a 5.08 cm Ge-etalon can be inserted between the laser and the first mirror. The light passes a 12 m optical path within the cylindrical multipass cell (89 reflections) before it leaves the cell through the same hole and is steered to the infrared detector. The cell is described

elsewhere in detail (Mangold et al., 2016) and only a brief summary is given here. As a cylindrical cell is always concentric in tangential direction, we chose the cell to be confocal in sagittal direction which leads to a more stable beam propagation in both directions compared to a concentric arrangement. Furthermore, the mirror curvature was chosen to be parabolic in sagittal plane for better refocusing properties of the beam. To minimize interference fringes an absorption mask with holes of 4 mm diameter, where laser reflections are expected, is inserted in the cell. Additionally, the beam is coupled into the cell in

an off-axis configuration leading to further separation of neighboring reflections on the mirror and thus less interference fringes. With a diameter of 14.5 cm the cell has a volume of 300 ml. In contrast to alternative multipass-cell concepts, like Herriott-cells, the cell is built from a solid ring. Therefore, it is less sensitive to internal misalignment due to shock or vibrations.

In order to obtain narrow, well separated absorption lines, the pressure in the cell is reduced to 100 hPa by continuously

pumping on it while the air flux is limited to 180 sccm by a 100 µm diameter orifice placed at the gas inlet. We determined

the flushing time for a complete gas exchange in the cell to be 9.6 s by exponentially fitting the $NO_2$ signal after switching from outside air to filtered $NO_2$ free air.

The detector signal is digitized by a digital oscilloscope (Picoscope 4000 Series, Pico Technology) at 12-bit resolution and with 20 MS/s sampling rate. The data acquisition is triggered by a TTL trigger signal generated by the laser driver. After acquisition of 1000 spectra at a rate of 2 kHz the data is transferred via a USB 2.0 port to a nano PC (Zero pro, Xi3 corporation) where the spectra are averaged and analyzed by a custom-written LabVIEW program. The $NO_2$ absorption spectrum is fitted with pressure, temperature, path length and line strength (HITRAN database (Rothman et al., 2013)) as input parameters to determine the $NO_2$ concentration according to Beer-Lambert's law. Pressure and path length are considered constant, while the temperature of the cell is measured with a thermistor and fed back to the fitting program. The chosen $NO_2$ absorption feature at 1599.9 cm$^{-1}$ offers a large dynamic range from 0 to several ppm which is largely sufficient for ambient concentration measurements in cities.

Since the instrument is exposed to the temperature variations of the outside air without having active temperature control, we have to regularly compensate for instrument drifts. This is done on two levels: First, the set-point of the laser temperature is adjusted via a feedback loop to lock the laser frequency, and thus the $NO_2$ absorption feature is kept at the same position in the measured spectrum. This measure counteracts the instrument drift by regulating the laser temperature. Second, the instrument repeatedly determines the zero point offset, i.e. the instrument signal at zero $NO_2$, by measuring filtered $NO_2$-free air for 2 minutes every 20 minutes. This zero air is obtained by pumping outside air through a filter of 20 ml volume filled with Chemisorbant media (Purafil, Inc. USA).

Via a serial interface, the $NO_2$ concentration data (3 s averages) are transferred to a measurement unit of the "OpenSense" network. This unit is installed next to our instrument and described in detail by Hasenfratz et al. (Hasenfratz et al., 2015). The position of the tram is determined by GPS. $NO_2$ concentrations and GPS positions are transmitted via GSM (Global System for Mobile communications) and stored in a database.

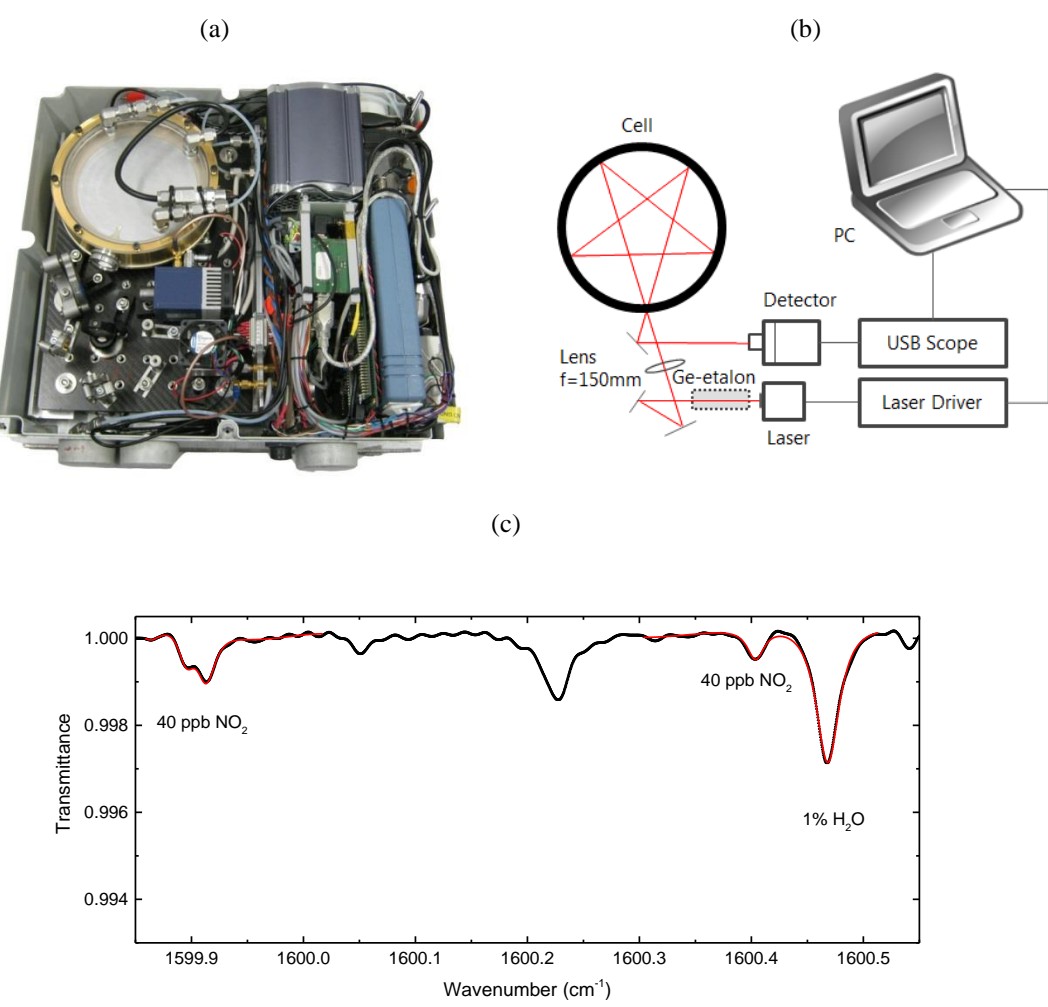

(a)

(b)

(c)

**Figure 1: (a) Photograph of the QCLAS instrument without the cover. (b) Schematic of the instrument and its optical layout. (c) Absorption spectrum recorded by the instrument. The red lines show the areas that were fitted to determine concentrations.**

## 2.2 Mobile $NO_2$ concentration measurements

5    The QCLAS instrument was placed on the roof of a tram of the public transport company of the city of Zurich (VBZ) from 21 Sept 2015 to 09 Mar 2016. This tram ran most of the time on the tram services no. 11 and 14 according to the operation schedule defined by VBZ (see supplementary materials for a detailed map). Tram services in Zurich operate from 5 a.m. to 1 a.m. at varying intervals. Consequently, the set of daily measurements (time period, number and location) varies from day to day. The cross-city tram services no. 11 and 14 link areas between 400 and 520 m above sea level while elevated residential

10    areas in Zurich are up to 640 m a.s.l. The routes of these services include short passages that are free of motorized traffic but mostly are on regular roads with little to heavy traffic.

We analyzed the $NO_2$ data from the period 09 Oct 2015 to 15 Feb 2016. Time periods for which high quality measurements are available are depicted in Figure 2. The gap between 27 Dec 2015 and 05 Feb 2016 is due to discarded data of reduced quality related to misalignment in the optical system of the instrument. Further, we excluded data when the tram was located within the depot area or when the GPS position was not clearly attributable to the correct tram track. Moreover, data were

omitted when the $NO_2$ spectrum was not clearly identified by the processing algorithm. The water concentration determined from the measured spectrum was used as an indicator to flag such data. In total 37 % of the measurement days yielded data that could be used for statistical modeling.

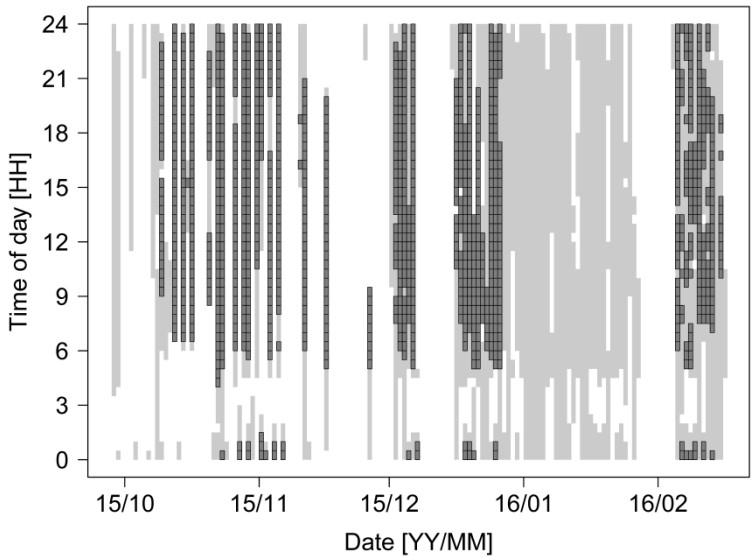

Figure 2: Time periods when the QCLAS instrument was in operation and $NO_2$ data were transmitted to the database are depicted
in light gray. Thirty minutes periods when the QCLAS instrument obtained high quality $NO_2$ data and the tram was in regular operation are depicted in dark gray (total: 1183). Statistical models were developed for these periods.

## 2.3 $NO_2$ concentration measurements at fixed sites

The municipal (Department for Environment and Health Protection (UGZ), City of Zurich) and federal (Federal Office for the Environment, FOEN) authorities operate seven air quality monitoring (AQM) stations with CLDs for regulatory purposes
in the city of Zurich. The set of stations includes background as well as highly polluted locations (see supplementary materials). It provides a comprehensive overview of the pollution situation encountered in Zurich with respect to the range of concentration levels and thus to the intra-urban variability. However, the spatial representation is coarse with respect to the generation of spatially highly resolved pollution maps. We had access to the one minute measurements from all sites for the time period 1 Jan 2015 to 1 June 2016. Two of the stations, STA and SCH, are located next to the tram tracks at a distance of
8 and 20 m, respectively. Five stations were operational during the complete campaign period. Operation of station BLU started on January 1, 2016. Station SWD was closed on February 1, 2016. The reported annual mean $NO_2$ concentrations in 2015 for the AQM sites were: HEU: 18 µg/m$^3$, ZUE: 31 µg/m$^3$, STA: 33 µg/m$^3$, SCH: 45 µg/m$^3$, SWD: 47 µg/m$^3$, RGS: 50 µg/m$^3$(OSTLUFT, 2016).

## 2.4 Spatial data

We employ spatial data for the classification of locations in the city of Zurich with respect to the instantaneous $NO_2$ concentration. $NO_2$ concentrations encountered in Zurich are moderate compared to other cities. Largest emission sources contributing to ambient $NO_2$ concentrations in Zurich are motorized traffic (47% of $NO_x$) and heating systems (28% of $NO_x$) (followed by industry (15%), constructions (9%) and others (1%)) (Brunner and Scheller, 2014). There is no heavy industry. Traffic emissions are by far dominant with respect to the spatial variability measured by our mobile instrument as the release points of the emissions from heating systems are usually well above street level.

We used average daily traffic volumes on particular roads, the digital elevation model, and building footprints and heights as spatial input data. The data was provided by the City and the Canton of Zurich and origins from the year 2013 which is well representative for the period observed in this study. From this data, we computed spatial variables representing the traffic intensity (several types), the "sky view factors" and the elevation above sea level following the procedures outlined in detail in Mueller et al., 2015 (Mueller et al., 2015) and Mueller et al., 2016 (Mueller et al., 2016). These variables cover the municipal area of Zurich in a 10 m grid.

Traffic intensity at a specific location was computed by summing up the distance travelled by vehicles in the vicinity of this location. The summands were weighted based on an exponential decay function which depends on the distance from the location. Moreover, we multiplied heavy vehicles by a factor of 10 in order to account for the higher emissions compared to light vehicles. We used four types of traffic intensities: the first attributes a weight of $1/e$ to summands at a distance of 30 meters (DTV030), the second at a distance of 50 meters (DTV050), the third at a distance of 100 meters (DTV100) and the fourth at a distance of 400 meters (DTV400). While the versions DTV030 and DTV050 depict the near field traffic intensity, versions DTV100 and DTV400 account for the far field traffic intensity.

The variable "sky view factor" (SVF) indicates the fraction of the sky that is visible at a particular location by means of the surface area of half of a unit sphere being related to the built environment.

The variable type elevation (DTM) indicates the altitude above sea level of the measurement locations.

## 2.5 Statistical modelling

Statistical models directly rely on measurements. Accordingly, the distribution and accuracy of the measurements strongly impact the model results. The QCLAS $NO_2$ instrument installed on top of a tram provides an average $NO_2$ concentration for every three seconds. While the tram is moving, numerous different locations in the city are probed. These measurements are complemented by those from the fixed AQM sites. The combined data set allows the investigation of two aspects that are important for statistical modeling:

First, we focused on small-scale features of the pollutant field. This subject was addressed by computing models for the prediction of the $NO_2$ concentration at the locations of the moving tram. Models relying on measurements from AQM sites allow quantifying how well pollutant concentrations at different places in the city can be predicted based on the existing infrastructure consisting of seven fixed locations. Models based on mobile measurements shed light on the potential of denser measurement networks such as mobile ones in cities like Zurich. Second, we analyzed for which parts of Zurich the tram measurements provide information about the instantaneous pollutant field. Reference measurements for model

validation are obtained from the spatially well-distributed AQM sites. Moreover, the application of the same modeling approach as used for the tram measurements to data from the AQM sites yields a benchmark for the performance of the mobile sensor network.

As outlined above, the modelling efforts are motivated by the request for accurate pollutant concentration maps. This study does not allow the generation of a long-term series of maps but adds knowledge of their generation.

An approach based on regression trees was developed (Breiman et al., 1984) and applied. Our approach relies on spatial information and $NO_2$ measurements and the basic assumption that $NO_2$ observations in the urban environment are similar at locations with similar spatial features. The main strengths of such a statistical modelling approach are the moderate computational time and the fact that information about the impact of instantaneous activities of emission sources and of meteorology is implicitly covered by the measurements.

The use of regression trees for the spatio-temporal analysis of the mobile and fixed $NO_2$ observations requires that any location in the city is described in terms of multiple spatial features. The chosen spatial features are DTV030, DTV050, DTV100, DTV400, DTM and SVF forming the location dependent input variables. The $NO_2$ observations and the input variables are linked by the GPS positions. Regression trees are described in detail in (Breiman et al., 1984). The basis of the approach is a set of data units referring to an observation y (mobile or fixed $NO_2$ measurement at location $L_i$ on time $t_i$) and related values of the input variables j (i.e. the explanatory variables). The algorithm performs repetitive partitioning of the data set by means of the spatial features and the $NO_2$ measurements resulting in M location types, $R_1$, $R_2$, …, $R_M$. Figuratively, starting from the undivided data set a tree is grown that consists of splits and leaves (in our context: location types). Splitting variables j and split points s are found by aiming at reducing the variance of the $NO_2$ measurements referring to a location type by successively solving

$$\min_{j,s}\left[\min_{c_1}\sum_{x_i\in R_1(j,s)}(y_i-c_1)^2+\min_{c_2}\sum_{x_i\in R_2(j,s)}(y_i-c_2)^2\right]$$

The modeled $NO_2$ concentration $c_i$ of location type $R_i$ equals the average of the $NO_2$ concentration measurements being part of $R_i$. The final number of location types or leaves depends on the number of successful splits. Splits are only attempted if at least 75 observations exist in a location type and the split increases the explained variance by 0.01. A location type $R_i$ must contain at least 25 observations.

A location type $R_i$ is the result of the splits in the domain of the spatial features. Accordingly, it does not represent a contiguous spatial area in reality but patches distributed in the city. $NO_2$ concentrations can be predicted for any location in the city based on the spatial features of these locations and the regression tree. We used the R package "rpart" (Therneau et al., 2015) for the computations.

# 3 Results and discussion

## 3.1 Instrument performance and stability

The precision of the instrument was determined using the Allan-variance technique (Werle et al., 1993). Figure 3 shows an Allan-deviation plot for measurements of filtered ($NO_2$-free) air in the laboratory. The best precision of 30 ppt is reached after 200 s averaging while the 1 s precision is about 300 ppt. To get some insight into the source of the instrument noise, we performed a dark-noise measurement. In such an experiment the spectral fitting algorithm is applied to the detector output while the laser light is blocked. The resulting Allan-deviation for dark-noise is nearly equivalent to the Allan-deviation of the zero-air measurement, indicating that the $NO_2$ precision is basically limited by the dark-noise of the detector until the minimum is reached. While the Allan-plot for zero-air levels off at about 200 s, due to instrument drifts, the noise of concentrations retrieved from dark-noise can be further reduced by longer integration. Such behavior is expected in case of the white noise limited regime. The Allan-deviation of zero-air measurements shows the excellent stability of the instrument as it remains at a low level even after 1000 seconds. Based on these results, the measurement strategy was designed such that it applies a drift correction to the concentration measurements by measuring $NO_2$-free air for 2 min after 18 min of measuring. This offset-correction is achieved by subtracting the linear interpolation of the mean of the zero-air measurements before and after each 18 min measurement period.

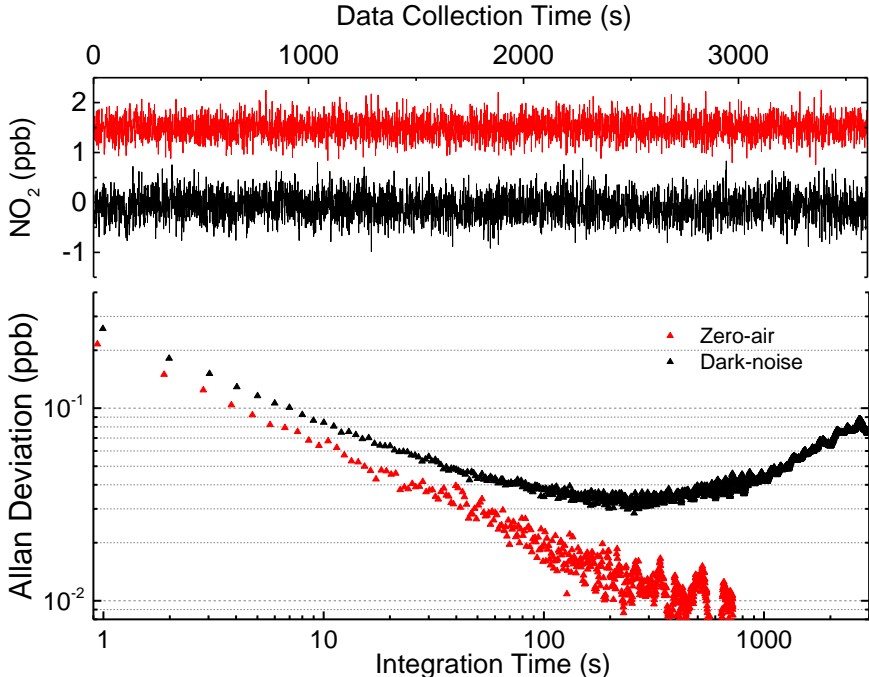

**Figure 3: Determination of the instrument precision and stability: Time series of zero-air (black) and detector dark-noise (red) and the associated Allan-deviation plots. The dark-noise trace is offset by 1.5 ppb from zero for clarity.**

The instrument accuracy was determined by comparing its retrieved values with data from the chemiluminescence detector (CLD, APNA 370, Horiba) of a fixed air quality monitoring site (NABEL, located in Dübendorf, Switzerland). The CLD has a limit of detection of 0.1 ppb and at 30 ppb a total uncertainty of 1.25 ppb  (NABEL, 2016). For this comparison, the instrument was installed on the roof of the air quality monitoring site next to the gas inlet of the CLD. A correction factor of

5  1.2 to the QCLAS data was found by plotting the QCLAS data over the corresponding CLD data. Such a correction to the normally calibration free QCLAS method was necessary because of loss of $NO_2$ during the sampling and in the long path cell and therefore was applied to all concentrations measured with the QCLAS. The contact of $NO_2$ with metal or plastic surfaces leads to dissociation. Since the pressure and flow rate in the cell are constant, the fraction of dissociated $NO_2$ can also be assumed as constant. Note that formation of $NO_2$ in the sampling line of instruments due to reaction of NO with $O_3$ can lead

10  to an overestimation of $NO_2$. The influence of this artefact is small, especially because the measurements have been done during the cold season at low $O_3$ concentrations. It is, however, for the QCLAS a bit higher than for the reference CLD instrument, because the residence time of the air sample is for the CLD somewhat shorter (about 6s).

The concentration measurements of both instruments after scaling of the QCLAS data are depicted in Figure 4. They yield excellent agreement over the full range of concentrations that was encountered during the 48 h period that is displayed

15  (17 June 2015 to 19 June 2015). Discrepancies between the two instruments are mostly due to the higher temporal resolution of the QCLAS. The standard deviation of the QCLAS measurements from the CLD was found to be 0.96 ppb therefore we conclude that the instrument accuracy is about 1 ppb with respect to the CLD and that the assumption of a constant relative loss of $NO_2$ was correct.

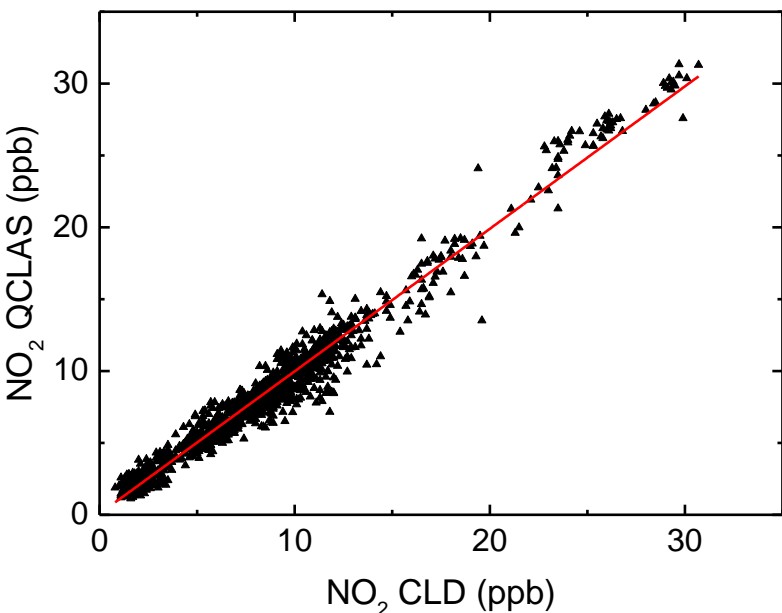

**Figure 4: Scatter plot of QCLAS concentration measurements (1 minute means) versus CLD measurements (1 minute means). Both instruments ran in parallel at the AQM site in Duebendorf, Switzerland, in the period 17 Jun 2015 to 19 Jun 2015. The red line is a linear fit to the data with slope 0.99±0.005, intersect 0.08±0.05 and an $R^2$ of 0.97.**

### 3.1.1 GPS positioning and map matching

The measuring unit on the tram was equipped with a GPS receiver providing a position every three seconds. GPS positions were orthogonally projected on the nearest tram track after the route of the tram had been determined. We discarded all the positions with corrections exceeding 10 m.

$NO_2$ measurements and GPS positions were not synchronous. Positions of the $NO_2$ measurements were linearly interpolated from the matched GPS positions. Hereby, we added 1.5 seconds to the timestamps of the matched positions. Such a delay is necessary because at a given time-stamp the GPS provided the present position while the $NO_2$ concentration is the average of 3 seconds before a time-stamp.

The $NO_2$ measurements represent 3 s averaged data, which, considering the traveling speed of the tram, cover a given route segment instead of representing a fixed point measurement. The length of this segment is the product of the integration time of 3 seconds and the speed of the tram. Tram speed associated with the mobile $NO_2$ measurements was smaller than 5 m/s for 64 % and smaller than 10 m/s for 92 % of the time.

### 3.1.2 Characteristics of tram based $NO_2$ measurements

Operation conditions during the mobile application are significantly more demanding than in the laboratory due to the harsh environmental conditions involving large temperature variations, vibrations, changing humidity and precipitation. We

analyzed the quality of the data obtained in the mobile application in two ways: First, we quantified the measurement uncertainty related to noise and variations of the zero point offset. Second, we compared the tram measurements to measurements from a fixed air quality monitoring site (see next section).

Figure 5 (a) depicts the $NO_2$ zero point offsets determined during the 2 minutes periods when $NO_2$-free air is pumped into the measuring cell (the cell is flushed during the first 45 s of the 2 minutes period; 25 measurements (75 s) remain to determine the zero point offset). $NO_2$ offsets outside the range [-10..10] ppb and $NO_2$ offsets derived in zeroing periods with less than 5 measurements or with a standard deviation exceeding 5 ppb were discarded. 88 % of the remaining zero point offsets are within ± 5 ppb. The corresponding standard deviation of a single measurement during the zeroing period is in the order of 0.5 ppb (Figure 5 (b)). Measurements that are not enveloped by two zeroing periods within a time interval of less than 20 minutes cannot accurately be adjusted for the zero point offset and were rejected.

We corrected the $NO_2$ measurements by linearly interpolating the zero point offsets derived in two consecutive zeroing periods. An upper limit for the interpolation error was derived by predicting the zero point offset of a 2 minute zeroing period by linear interpolation of the two neighboring zero point offsets (40 minutes time span). The resulting standard deviation of the difference between predicted and measured zero point offset amounts to 1.5 ppb (Figure 5 (c)). Zero point offsets are temporally correlated as the standard deviation of the differences of two consecutive zero point offsets amounts to 2.1 ppb and thus is smaller than 4.3 ppb expected for white noise ($\sqrt{2} \cdot \sigma_{NO2\ ZPO}$).

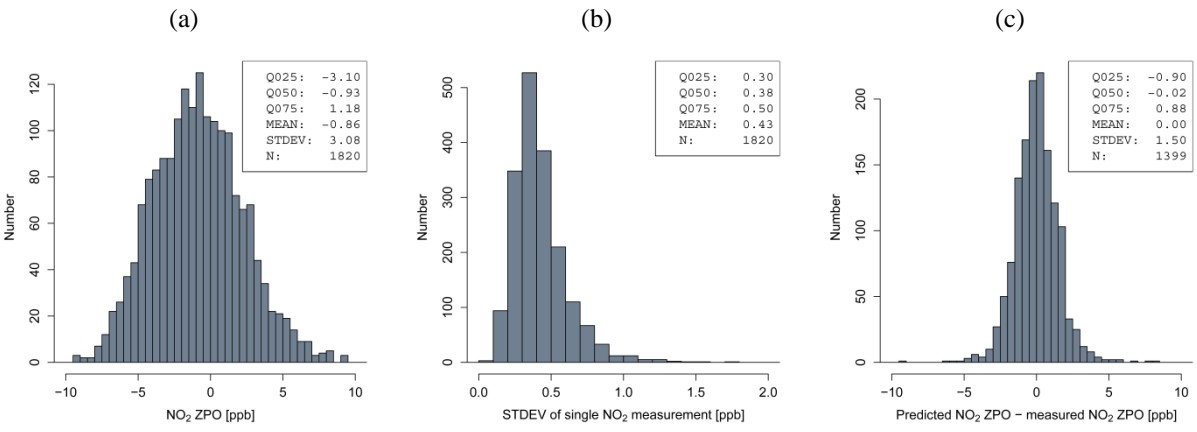

**Figure 5: (a) Histogram of $NO_2$ zero point offsets (ZPO) computed during the calibration periods. Q025, Q050 and Q075 denote the 25%, 50% and 75% quantiles, respectively. (b) Histogram of the standard deviations of a single measurement during the zero point measurements. (c) Differences between predicted and measured zero point offsets.**

## 3.2 Intra-urban and temporal variation in $NO_2$ concentration

The measurements of the mobile QCLAS instrument reveal the spatio-temporal variation of the urban pollutant concentration field. As an example, Figure 6 shows one day time series of tram measurements (3 seconds and 5 minutes averages) and of measurements (5 minute averages) from the fixed monitoring stations SCH and STA. The fixed air quality monitoring stations STA and SCH are located at 8 and 20 m distance from the tram tracks, respectively, and are equipped with CLDs measuring $NO_2$. The tram passed the fixed sites about every 30 minutes on the selected day. Site SCH is

impacted by heavier traffic than site STA resulting in higher NO$_2$ concentrations on average. Annual mean concentrations at SCH and STA in 2015 were 23.6 ppb and 17.3 ppb. Traffic in the morning and evening rush hours in Zurich is comparable. Hence, a large part of the diurnal variation in NO$_2$ concentration is due to meteorology, e.g. the development of the boundary layer height during the course of the day.

The tram carrying the QCLAS instrument regularly passed AQM sites STA and SCH during the measurement campaign. This provides the opportunity to directly compare the measurements from the tram (3 seconds averages) and the measurements from the fixed site (1 minute averages). We plotted single measurements from the mobile instrument that were obtained in a 30 m radius around the fixed station STA (a) and SCH (b) over the corresponding 1 min average values from these AQM stations in Figure 7. The comparison reveals excellent agreement of measurements from the QCLAS

instrument to measurements from site STA, whereas the agreement between the tram measurements and the measurements from station SCH is less pronounced. This can be expected for two reasons: (i) the larger distance, (ii) the location of site SCH at a crossroad where traffic flow is controlled by traffic lights and where the short term variability of NO$_2$ is larger. From a more general perspective, this is of importance when mobile sensor data is corrected based on data from nearby fixed reference sites, as suggested by (Arfire et al., 2015; Saukh et al., 2015). The parameters for data correction as determined

from such an approach depend on the local concentration variability and the time response of the sensor and the reference instrument.  Probing different air parcels may concur with concentration differences as it is observed for the data set obtained for AQM site SCH. Parameters determined from such a data set could be associated with significant errors depending on the temporal stability of the sensor, the applied mathematical sensor model and the term used in the mathematical model for the description of the concentration differences.

Figure 8 depicts the NO$_2$ measurements on the track section from station "Bellevue" to "Bahnhofquai" for the full measurement campaign (148 runs) with 50 meter resolution. Figure 8 shows the small-scale NO$_2$ variations that are measured by the mobile instrument. At sections where the tram is operating on congested roads (between the first and the third stop as well as between the fifth and the seventh stop), the average NO2 concentration is higher than at sections with very little traffic. Moreover, Figure 8 shows the spatial distribution of the number of measurements taken, which is highest

close to tram stations where the velocity of the tram is low or zero.

Our statistical models are driven by measurements and are based on the assumption of comparable NO$_2$ concentrations at locations that are similar in terms of vicinity to sources (in particular traffic), the built environment and elevation. They cannot provide accurate predictions for locations with NO$_2$ concentrations significantly higher or lower than observed. We analyzed this subject by a comparison of all the compiled tram measurements and the measurements from each AQM site (5

minutes averages, respectively). Table 1 shows that NO$_2$ concentrations measured on the tram are in line with concentrations measured at AQM sites (roadside, urban background) in the city center. They are most similar to the measurements from the AQM site STA (RMSE: 7.8 ppb, slope: 1.10, Δ mean=3.2 ppb). Site STA is most representative for the set of location types passed by the tram. In contrast, NO$_2$ concentrations at the elevated (200 m above the city center) background site HEU are rarely in the range of the tram measurements, demonstrating the limited representativeness of the tram-based data set for the

entire city.

**Table 1: Comparison of NO$_2$ tram measurements and measurements from the AQM sites, aggregated to 5 minutes means. N denotes the number of measurements, r is the Pearson correlation coefficient, and slope refers to a regression line through the**

origin (tram against AQM), Δ mean is the difference between the average concentrations (tram minus AQM) and Q005, Q050 and Q095 denote the 5%, 50% and 95% quantiles of the differences tram minus AQM.

| AQM site | N | RMSE [ppb] | r | slope | Δ mean [ppb] | Q005 [ppb] | Q050 [ppb] | Q095 [ppb] |
|---|---|---|---|---|---|---|---|---|
| BLU | 1148 | 11.6 | 0.72 | 1.33 | 8.0 | -4.3 | 7.1 | 22.3 |
| HEU | 6608 | 15.4 | 0.58 | 1.65 | 12.4 | 0.4 | 10.9 | 29.3 |
| RGS | 6611 | 10.4 | 0.57 | 0.86 | -3.4 | -20.9 | -2.4 | 10.3 |
| SCH | 6220 | 9.5 | 0.67 | 0.88 | -2.4 | -17.6 | -2.0 | 11.5 |
| STA | 6649 | 7.8 | 0.76 | 1.10 | 3.2 | -7.3 | 2.7 | 14.8 |
| SWD | 5470 | 11.3 | 0.46 | 0.85 | -3.1 | -20.7 | -2.8 | 13.9 |
| ZUE | 6558 | 9.5 | 0.72 | 1.22 | 5.6 | -5.2 | 4.9 | 18.5 |

The temporally highly resolved tram-based measurements provide information about the $NO_2$ concentration distribution along the track. We analyzed the distribution of the $NO_2$ tram measurements in 30 minutes modelling periods (Ti) by computing the 5% and the 95% quantile of the measurements as well as the difference between the 95% and the 5% quantiles (Figure 9 (a) and (b)). The difference between the 5% and the 95% quantiles, respectively, and the mean value remains rather constant for increasing mean values. This points out that temporal variation (i.e. the mean values of different modeling periods) in the tram based $NO_2$ measurements is of comparable magnitude to intra-urban variation (difference between the 95% and the 5% quantiles within a modeling period). The difference between the 95% and the 5% quantiles is below 35 ppb in about 90% of the modelling periods. Short-term variation in $NO_2$ concentration at a specific location is mainly caused by changing activities of emission sources in the close vicinity. This is observed at all AQM sites in Zurich that are impacted by traffic (Figure 9 (c) and (d)) and can be of similar magnitude than the $NO_2$ variations measured by the tram in motion.

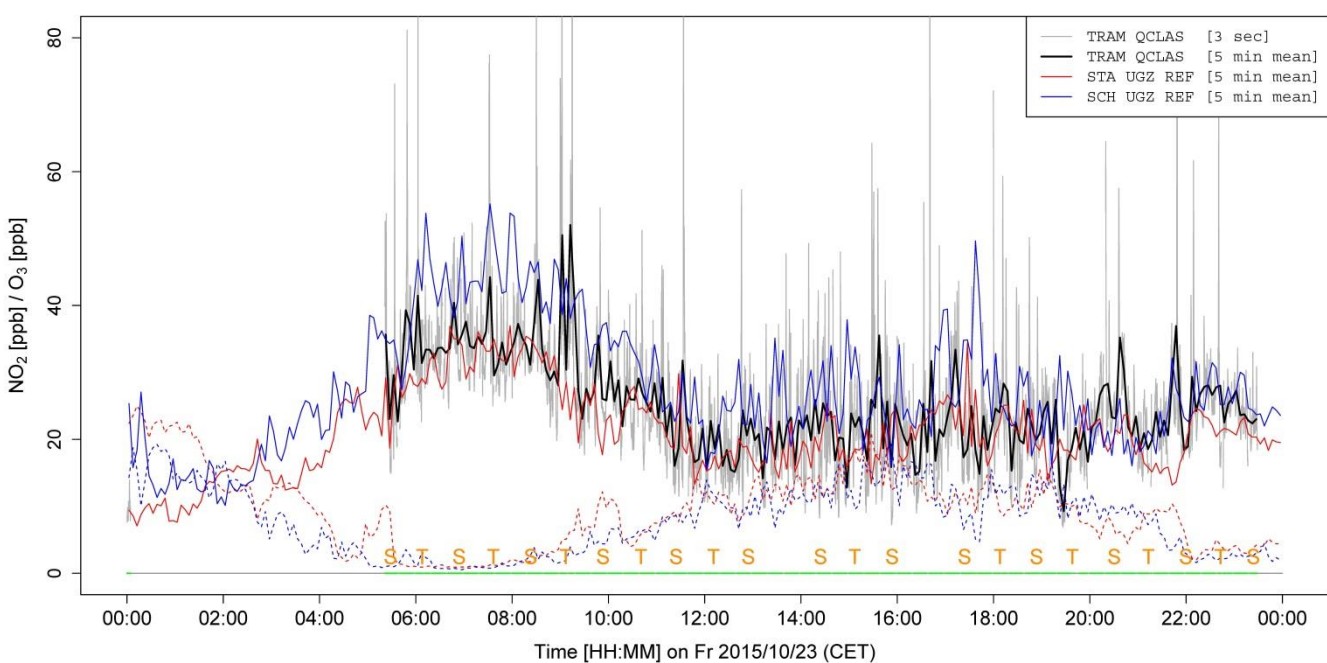

**Figure 6:** Example time series of measurements of the tram based QCLAS instrument (single measurements and 5 minutes means) as well as of NO₂ and O₃ measurements from the AQM stations STA and SCH (5 minutes means). The letters "S" and "T" depict when the tram passes the terminal stations Seebach and Triemli.

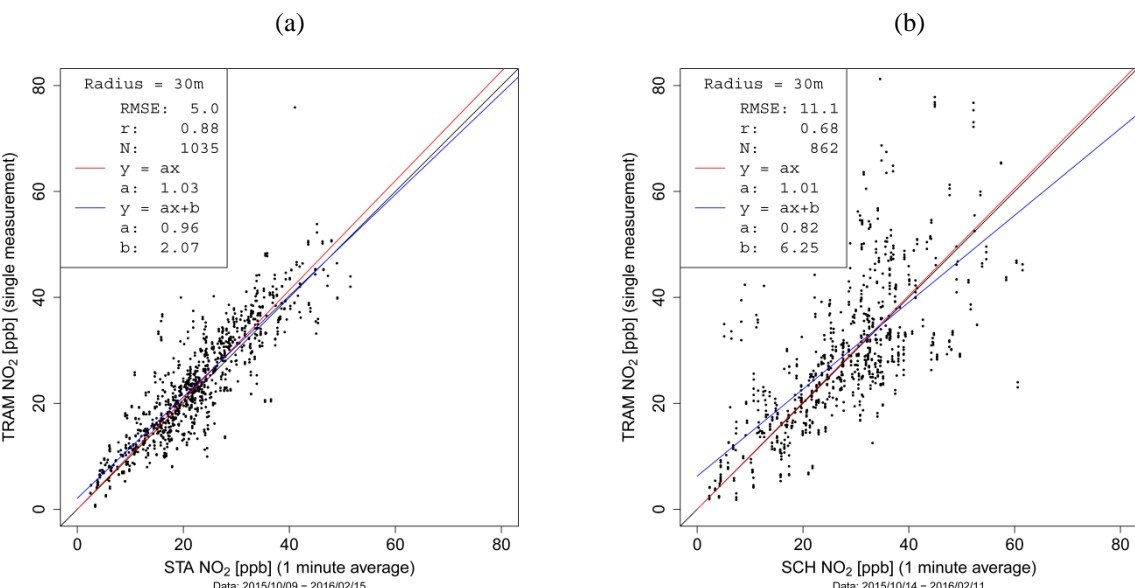

**Figure 7: Comparison of measurements from the QCLAS instrument (single measurements) and measurements (1 minutes mean) from the AQM sites STA (a) and SCH (b) when the tram is within a 30 meter radius from the respective AQM site. RMSE is the root mean square, r denotes the Pearson correlation coefficient and N is the number of 3 seconds measurements from the tram.**

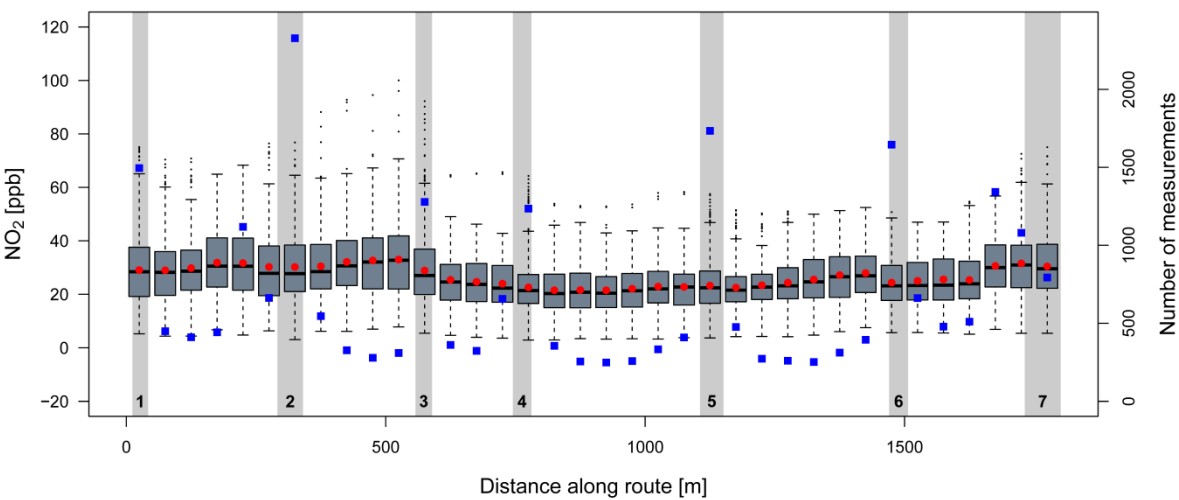

Figure 8: Boxplot of QCLAS measurements taken on all runs (n=148) between station "Bellevue" and station "Bahnhofquai" during the entire campaign and to 50m track length. The red dots indicate the $NO_2$ concentration means of the boxes, the blue rectangles depict the number of measurements referring to particular boxes, the black dots are measurements outside the whiskers. Most measurements were taken near tram stops (depicted in gray, black numbers). The tram is directly impacted by traffic between the first and the third stop as well as between the fifth and the seventh stop. There is only little traffic between the third and the fifth stop.

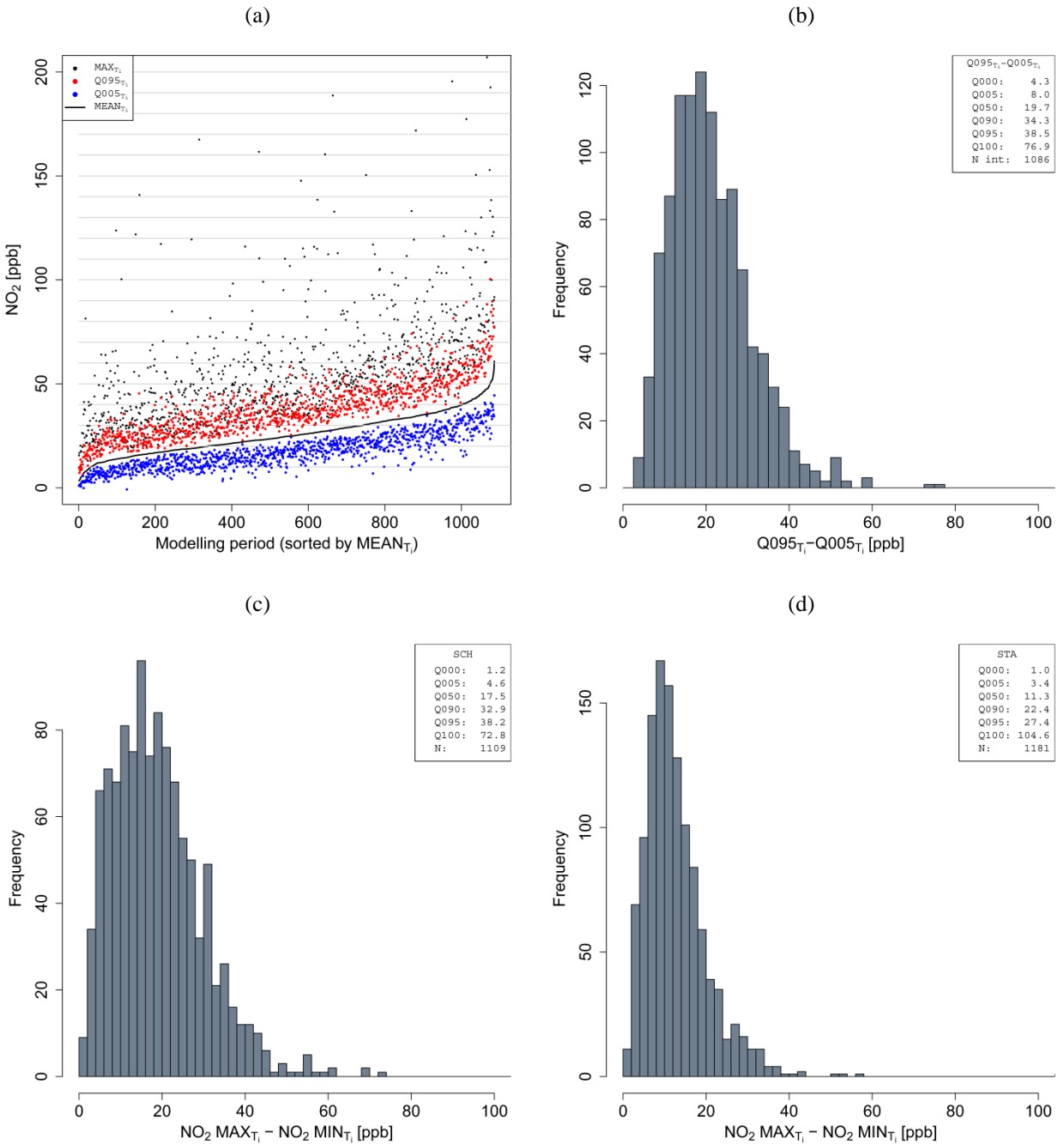

**Figure 9:** (a) 5% quantile ($Q005_{Ti}$; blue dots) and 95% quantile ($Q095_{Ti}$; red dots) of the $NO_2$ measurements (3 seconds) as well as the maximum (black dots) and the mean (black line) in each modelling period $T_i$ (30 minutes duration, approximately 500 measurements; periods $T_i$ with less than 300 measurements were not used for the plots in (a) and (b)). The intervals are ordered by the mean. (b) Distribution of the differences between the 5% and the 95% quantiles of the measurements in the modelling periods $T_i$. (c) Histogram of the differences between the maximum and the minimum 1 minute $NO_2$ measurements from AQM site SCH in the modelling periods $T_i$ (30 minutes). (d) Same as in (c) for AQM site STA.

### 3.3 Results of statistical modelling

We applied the procedure described in section 2.5 to selected sets of $NO_2$ measurements obtained by the mobile QCLAS instrument and the fixed AQM stations. Independent regression trees are computed for each 30 minutes period. Setting the modelling period to 30 minutes is a compromise between the number and spatial distribution of mobile measurements

forming the input for the regression tree computation and the temporal variation of the $NO_2$ concentration, i.e. changes in emission source activity and meteorology which may alter the relation between covariate values and $NO_2$ concentration.

The locations within the municipality of Zurich were described by means of six spatial variables (i.e. explanatory variables). Various combinations of these variables were defined, and regression trees relying on these variable selections were computed in order to find the best performing model for each particular setting during all modelling periods.

The number of location types for which representative $NO_2$ concentrations can be determined based on our statistical models is limited. The number of measurements required for estimating an accurate mean concentration for a location type depends on the short-term temporal variability of the $NO_2$ concentration (see Figure 9 (c) and (d) and section 3.3.2). However, the number and spatial distribution of the measurements in a modelling period (approximately 500) is given by the tram route and cannot be influenced. Accordingly, the use of too many explanatory variables in our statistical models may lead to an

over-determined model. Regression trees based on tram measurements have about 6 location types $R_i$ on average.

Models based on a single variable perform best when relying on a traffic related variable. Thus, elevation and sky view factors are of secondary importance. Predictions based on a basic model can easily be validated by cross-validation and by comparison with measurements from fixed AQM sites.

### 3.3.1 Model validation by means of three seconds $NO_2$ measurements from the tram

Three seconds mean $NO_2$ concentration referring to time and location of the tram were predicted in three versions. In version $V_T$, the data of the 30 minutes interval were partitioned into six five minutes intervals, denoted as A to F. Three independent regression trees were determined based on the tram measurements from BCEF, ACDF and ABDE and model predictions for AD, BE and CF were computed, respectively. In version $V_{AQ}$, regression trees based on the measurements from the fixed air quality monitoring sites were computed. In version $V_{STA}$, 30 minutes means of the measurements from the fixed site STA

were taken as predictions for the tram measurements.

The tram measurements as well as the computed predictions were aggregated to 5 minutes averages for being used in the statistical analysis. The predictions based on the tram measurements (version $V_T$) are best followed by those based on measurements from the fixed sites (version $V_{AQ}$) and those based on measurements of site STA (version $V_{STA}$). The difference in performance between $V_{AQ}$ and $V_{STA}$ is small due to the similarity of $NO_2$ concentration encountered at site STA

and by the tram-based instrument. The results are summarized in Table 2.

**Table 2: Summary of the comparison of model predictions and measurements from the tram (5 minutes means, respectively). r is the Pearson correlation coefficient, slope denotes a regression line through the origin (predictions against observations), N denotes the number of 5 minutes values. The variables used in the relevant model are listed in the last column.**

| Setting | RMSE | r | Slope | N | Used variables |
|---------|------|------|-------|------|----------------|
| $V_T$ | 5.6 | 0.86 | 0.97 | 6584 | DTV050 |
| $V_{AQ}$ | 7.3 | 0.78 | 0.89 | 6599 | DTV030/DTM |
| $V_{STA}$ | 7.6 | 0.79 | 0.84 | 6588 | - |

### 3.3.2 Model validation by means of 30 minutes mean $NO_2$ concentrations at fixed AQM stations

We computed a set of regression trees using different sets of measurements for each 30 minutes modelling period in order to predict $NO_2$ concentrations at the AQM sites. Version $W_T$ is based on tram measurements only, version $W_{AQ}$ is based on the measurements from the fixed AQM sites only (excluding the measurements from the target AQM site, respectively), and version $W_{T+A}$ is based on the tram measurements and the measurements from the fixed AQM sites (excluding the measurements from the target site). We attributed a weight of 20 to the AQM site measurements with respect to the QCLAS

measurements in version $W_{T+A}$. Models of version $W_T$ rely on the variable DTV100, models of versions $W_{AQ}$ and $W_{A+T}$ rely on the variable DTV030. Model versions using these variables provide the best overall fit to the measurements of all AQM sites. In version $W_B$ we simply use the 30 minutes mean $NO_2$ concentration measured by the tram as prediction for the concentration at respective AQM sites as benchmark for versions $W_T$, $W_{AQ}$ and $W_{T+A}$.

A series of three maps illustrates differences between the versions $V_{AQ}$, $V_T$ and $V_{T+A}$ in terms of the measurement

distribution and the resulting $NO_2$ concentration field (Figure ). For optimal comparison, all these maps are based solely on the variable DTV050. Concentrations are shown only for locations with DTV050 values within the DTV050 range covered by the measurements as extrapolations based on statistical models are associated with larger uncertainties. Obviously, combining the measurements from AQM sites and from the tram yields the data set with the largest spatial coverage. The figure shows that the spatial coverage is not optimal, yet, in any version. Considerable potential remains in the optimization

(i.e. extension) of static or mobile measurement networks.

Figure  depicts a set of indicators for the agreement between model predictions ($W_T$, $W_{AQ}$, $W_{T+A}$ and $W_B$) and 30 minutes mean concentrations measured at the AQM sites. The results show that the classification approach in version $W_T$ reveals reasonable, bias free predictions for sites that are similar to locations passed by the tram (slope: 0.93-1.18, differences of average concentrations $\Delta_{mean}$: -1.1-4.8 ppb). The scattering in version $W_T$ is not smaller than in version $W_B$ but this can partly

be related to the partitioning of the tram measurement set into m location types $R_i$ by the classification approach.

In our statistical models, the mean of all the measurements in $R_i$ (covering approximately 1/m of 25 minutes with mostly m = 6) is taken as the estimate value for the 30 minute mean concentration of location type $R_i$. We performed the following computation in order to analyze the agreement between the mean derived from a limited number of measurements and the true 30 minute mean concentration for several locations. For this analysis, measurements from single AQM sites from the

year 2015 were used. We computed 5 minute means and complementary 25 minute means (6 pairs in 30 minutes,

respectively). The obtained RMSE values range between 1.5 ppb at HEU (elevated background site) and 7.5 ppb at RGS (roadside). Therefore, differences between the mean of a subset of tram measurements referring to a particular location type and its 30 minute mean have to be expected if $NO_2$ concentration at this location is highly variable on the short-term.

The results of model version $W_{AQ}$ are comparable to version $W_T$ which points to a good spatial distribution of the monitoring stations. The combination of measurements from AQM sites and from the tram (Version $W_{T+A}$) performs best as this model version, in contrast to $W_T$, also includes information about the urban background.

The tram instrument is rarely situated in background locations. Hence, $NO_2$ concentrations are overestimated by $W_T$ at AQM sites representing these types of locations (BLU, HEU and ZUE). To overcome this limitation, $NO_2$ concentration measured by the tram instrument might be considered as resulting from a background concentration plus an increment depending on the activity of nearby emission sources. Accordingly, urban background concentrations can also be estimated based on a baseline instead of the mean of a particular set of measurements. We defined the urban $NO_2$ concentration baseline as the 20% quantile of the $NO_2$ concentration measurements from the tram instrument in 30 minutes. Comparison of these quantile values with the means of the urban background site ZUE results in RMSE, Δmean and slope of 5.1 ppb, 0.5 ppb and 1.00, respectively. The baseline approach therefore outperforms version $W_T$ for site ZUE.

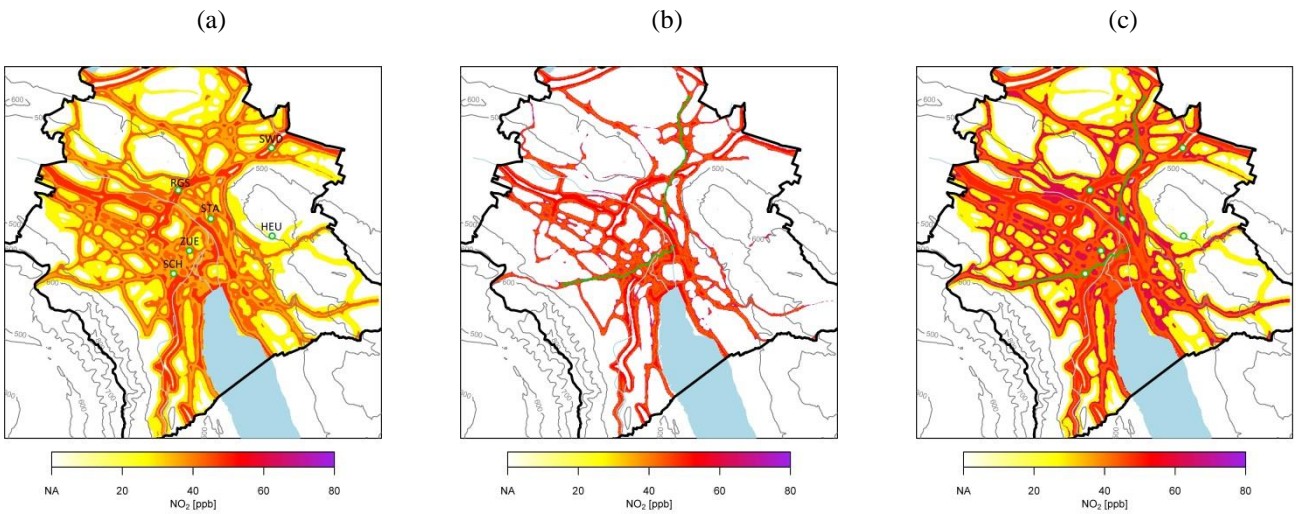

Figure 10: Series of $NO_2$ concentration maps for 17 Dec 2015 18:00-19:00. The models are based on the variable DTV050. Measurement locations are depicted by green dots, the tram track by a green line. (a) Map based on data from AQM sites (Version $V_{AQ}$). (b) Map based on measurements from the tram based QCLAS instrument (Version $V_T$). (c) Map based on measurements from the AQM sites and the QCLAS instrument (Version $V_{T+A}$). The maps cover a 12 × 12 km area. Light blue lines and areas depict rivers and lakes, the thick black line shows the border of the municipal area of Zurich and the gray lines indicate the contour lines of elevation (500 to 800 m a.s.l.).

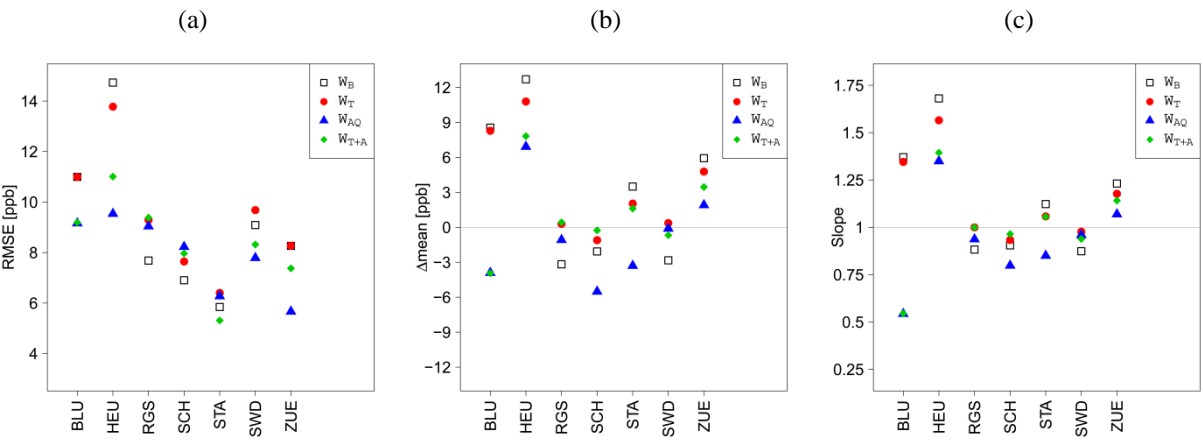

**Figure 11: Comparison between observed 30 minutes means at AQM sites and model predictions based on the versions $W_B$, $W_T$, $W_A$ and $W_{T+A}$. RMSE denotes the root mean square error, $\Delta$mean equals $1/N \cdot (\sum predictions - \sum observations)$ and slope refers to a best-fitting regression line through the origin (predictions over observations). N is 222 for site BLU, 955 for site SWD and between 1109 and 1183 for the other sites.**

## 3.4 Discussion and outlook

The mobile QCLAS instrument provided accurate $NO_2$ measurements of high spatial resolution but with sparse sampling for particular location types related to the accessibility of the measurement locations that were limited to the tram tracks (few background locations, no elevated locations). In principle, this kind of concentration data set can be improved for statistical modelling by the operation of a larger number of instruments. Options are the installation of additional instruments on further trams and on other public transport vehicles (e.g. buses connecting residential areas running on roads with very low traffic) or the operation of instruments at fixed background sites or at highly polluted locations. Due to the number of required measurements for location types with highly variable $NO_2$ concentration, the necessity for measurements at locations with different pollutant situations and the travel time from one to another location, we suggest that the number of required mobile or static QCLAS instruments for comprehensive $NO_2$ measurement in Zurich is not smaller than the current number of fixed AQM sites.

The models derived from tram measurements yielded the most accurate predictions for tram measurements. This is related to the small spatial distance between measurements and predictions as well as to the dense spatial distribution of the tram measurements. The latter corroborates the potential of dense measurement networks, either static or mobile ones, for obtaining more detailed information on the instantaneous pollutant field.

Mobile measurement devices that provide accurate long-term measurements cannot only contribute to map generation but also to the validation of products derived from pollutant maps. For example, the exposure estimations can be validated based on measurements of such instruments.

With the latest dual-wavelength QCL technology (Jagerska et al., 2014; Süess et al., 2016), the next generation of the QCLAS instrument could measure NO and $NO_2$, simultaneously, with the same compact geometry. Such measurements

would give valuable insights into urban $NO_x$ chemistry. Furthermore, other target species, such as CO, $CO_2$, $O_3$ or $CH_4$, may in the near future be included.

## 4 Conclusions

We presented a compact and robust QCLAS spectrometer that operated autonomously on the roof of a tram to measure $NO_2$
concentrations in Zurich over a period of four months (with an interruption of five weeks). The analysis of the measurements from periods when the instrument was operated next to a CLD of a fixed AQM site and from mobile operation on a tram showed that the accuracy of 3 second values in the field is better than 2 ppb with respect to the CLD that was used for initial calibration. Independence from roadside calibration infrastructure as well as from on-board calibration gases is a clear asset of the instrument for operation in urban environments with highly variable $NO_2$ concentration fields.

The QCLAS instrument is highly suited for mobile applications. The statistical modelling showed that the mobile measurements provided accurate information about the urban $NO_2$ concentration field. However, comprehensive analysis of the data from the tram and the AQM sites revealed large spatio-temporal variability in $NO_2$ concentration in the vicinity of emission sources. Accordingly, the main shortcomings of the obtained data set with respect to statistical modelling are the limited number of observations and the incomplete coverage of different types of locations. Improving the data set would
require the operation of a larger number of instruments. This would allow the mapping of the entire $NO_2$ concentration field in a city with high spatio-temporal resolution and, concurrently, the refinement of statistical modelling techniques.

### Acknowledgements

The support from the group of Computer Engineering & Networks Lab of ETH Zurich in the transfer and storage of the mobile $NO_2$ measurements is gratefully acknowledged. In this context, we thank Balz Maag for his help to install the
instrument on the tram. The Department of Environment and Health Protection (UGZ) of the City of Zurich, the Department for Waste, Water, Energy and Air (AWEL) of the Canton of Zurich and the Swiss Federal Office for the Environment (FOEN) is acknowledged for providing $NO_2$ concentration measurements from air quality monitoring sites and/or spatial data. We gratefully acknowledge the support of the transport company of Zurich (VBZ). The project was financially supported by nano-nera.ch (IRSENS II and Opensense II).

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
