# Peer review of "Mid-IR spectrometer for mobile, real-time urban NO2 measurements"

_Atmospheric Measurement Techniques, 2017_

## Referee Comment (RC1) · Anonymous Referee #2 · 28 Apr 2017

**Comments to manuscript AMT-2017-30**

The paper "Spectroscopic real-time monitoring of NO2 for city scale modelling", by Hundt et al, reports on mobile NO2 measurements performed in the city of Zurich with a QCL spectrometer placed on a tram roof. Data are analyzed and compared to AQM sites where CLDs measurements are available. A statistical data analysis is presented to provide NO2 concentration maps.

This work is interesting however it is hard for a reader to judge because some clue information are missing as listed in the general comments. The manuscript will need a major revision to be considered for publication in AMT.

**I) General comments :**

1) The need of mobile measurements is well introduced but an overview of available instruments is missing (commercial or laboratory prototypes). Here CLDs and electrochemical sensors are mentioned and references are given for instruments previously developed by the authors in the MIR. However there is no mention to the many different spectroscopy techniques used in the MIR as well as in the blue spectral region with ECDL or LED. Please compare the performances of the spectrometer developed with these previous works that have to be cited in this paper.

2) P2,L28 " The portable instrument was specifically built for this campaign". Figure 1 is just a scheme that could fit to all multipass spectrometer. The reader expects a detailed description of the technique (iCW mode) and the set-up to understand how the good performances reported in the following are reached. A spectra recorded by the instrument should be shown. Please, detail the "Quantum cascade laser spectrometer" section in this way (see detailed comments in the following).

3) The statistical model is based on average NO2 measurements during 3s (P6,L18- P9,L15). Note that the response time of the instrument is longer : 9.6s. Averaging at this time scale should be performed prior to analysis.

4) NO2 concentration is expected to be sensitive to the solar activity, NO, and O3 concentrations, and to ambient temperature and humidity. Variations of NO2 measurements should be studied according to these different parameters. It is hard to believe that all these parameter effects are removed by the offset correction.

5) QCLAS measurements is compared to CLDs measurements taken as reference values. Specification of the CLDs are expected : sensitivity, absolute accuracy, intercomparison of the CLDs...

**II)** Detailed comments :**

**a)** In the abstract and the introduction : please mentioned the technique used and the wavenumber of the laser.

**b) Introduction :**

1) This work is dedicated to urban monitoring, please specify the NO2 concentration range to clarify the instrument performances required in term of detection limit.

2) Insert references to previous NO2 spectrometers (see General comments)

3) Please gives accurate numbers concerning instrument performances :
P1, L30 "accurate" ?
P2, L9 : "not fast enough"?
P2 L11 "not short enough"

P2 L13 "relatively high" P2 L16 : "high sampling rates"

**c) 1.1 QCL spectrometer :**

1) The iCW operation was published in Fischer 2014, reporting on a different instrument that operates at 7.7 $\mu$ m. In this paper, a short description of this technique is expected and a detailed description of the optical set-up at 6.25 $\mu$ m as well. I could not find mention to an optical etalon : How is obtained the frequency linearization ?

2) Please show a recorded spectra around 1600 cm-1. How many NO2 lines do you monitor ? Is the spectra large enough to monitor water ?

3) In figure 1 : the detailed optical scheme would be more useful than the schematic content of the instrument

4) P3, L17 : please specify that the 100 $\mu$ m orifice is placed at the inlet of the cellule and give the corresponding flow value.

5)P3 L17-19 : The response time is a main concern. It is measured to be equal to 9.6s by switching from NO2 free air to NO2 sample. Is this really consistent with the complete exchange time of the cell according to the flow and pressure values ?

6) P3L22 : what is the acquisition rate ?

7)P3, L23-25: Concerning the fitting procedure : Please indicates which parameters are fixed (such as P probably) or left variable. The instrument not being temperature stabilized, what about the temperature parameter ? Is the path length deduced fixed?

8) Water absorption has strong absorption lines around 1600cm-1, and its concentration is changing during tram measurements. How is that taken into account ? Do you include a fit of water line to monitor it ? is it included in the base line?

9) P3,L29: please make explicit what you call "zero-point offset".

10) The dynamic range of the instrument should be given somewhere in the paper

**d)2.3. NO 2 concentration measurements at fixed sites**

Indicates that it is performed by CLD. Please give the specifications of these instruments (see general comments).

e) P5 L17 : " NO2 concentrations encountered in Zurich are moderate compared to other cities." Please specify here the typical range.

**f) 2.5 Statistical modelling**

1)P6, L18 : NO2 concentrations should be averaged to 9.6s, the response time (see general comments)

2)Please provide the reference (Breinam1984) concerning the "regression tree approachat first mention: P6, L32

**g) 3.1. Instrument performance and stability**

1) The Allan deviation is expected to increase at long integration time due to technical drifts or ambient parameters change. Can the author provide a longer set of measurements to show when it appears ?

2) In figure 3, in the upper panel can you confirm that Dark-Noise measurements are the upper one or is there a color inversion ? I would understand an offset of about 1.5ppb is the measured in" NO2 free air" rather than Dark-Noise, from a non perfect filtering or other effects.

3)P8, L1 : Concerning the "measurement strategy" I understand how the drift is corrected but this process relies on the zero NO2 gas production in-situ. Did you test that there is no NO2 trace in this gas ? I would expect some NO2 trace at the ppb level (according to fig 1). And the residual concentration is expected to depend on NO2 concentration in ambient air. Furthermore, the sensitivity to water concentration should be investigated . Can you provide some details ?

Another aspect should be detailed : how is calibrated the correction, on which reference measurement ? Did you performed a referenced measurement prior in-situ measurements ? Did you repeat it during the measurement campaign?

4) Allan deviation only demonstrate the stability of the instrument but not the accuracy. Caption of figure 3 is to be modified and 2 P7, L24-25 : "precision" should be replaced by "stability"

5) P8, L8-14 : I would expect the correction factor to change in time due to passivation of the surfaces or water interferences : the calibration performed in June is probably different during winter. Did you test this point ?

6) The correction factor of 1.2 is applied. But how was the instrument firstly calibrated ?

7)P9L2: Please specify the absolute accuracy of the CLD to justify the 1ppb accuracy of the instrument.

8) P9L15-19 : as mentioned previously, measurements should be averaged to 9.6s resulting in a longer route segment.

**h) 3.2 Intra-urban and temporal variation in NO 2 concentration**

1) Fig 6 : in the caption please clarify which AQM station (STA or SCH) is close to which terminal stations (Seebach or Triemli)

2)P11L7 : again measurements should be averaged to 9.6s.

3)P11,L15-17: I don't understand why "this analysis clearly shows the advantage of a mobile measurement device that does not rely on fixed sites for the determination of its calibration

parameters". I would assume that the main reason for the discrepancies observed with SCH measurements is due to the difference of averaging time : 9.6s response time for the QCLAS and 1min for the CLD. SCH being impacted by heavier traffic than STA , fast events of strong pollution will have more impact.

4) Figure 8 : What is the meaning of points after the 7th stop?

5) In this section, temporal variation of NO2 should be studied as a function of other parameters : NO, O3, H2O,T,solar activity (see general comments).

**i) Model validation by means of three seconds NO 2 measurements from the tram**

Should be performed with 9.6s averaging measurements

**j) 3.3.2 Model validation by means of 30 minutes mean NO 2 concentrations at fixed AQM stations**

1) Fig 10: change the color code of NO2 concentration to allow to distinguish between very low concentrations (presently in white) from the zone where the model cannot compute concentration. Presently, the spatial coverage is not clear.

2) P17, L29-30 : " The obtained RMSE values range between 1.5 ppb at HEU (elevated background site) and 7.5 ppb at RGS (roadside)." Given values are not consistent with Fig 11.

3) What is the error/shift value that can be attributed to the absolute calibration of the different CLDs ? Were they regularly calibrated on the same gas sample ?

**k) Conclusion**

 P20L2 : "QCLAS spectrometer that operated autonomously.... over a period of four months" should be added a mention to the intervention for optical alignment
 Please indicate clearly that the 2ppb agreement is obtained after a correction factor of 1.2 derived from these measurements
 3) 3s to be replaced by 9.6s.

**Typesetting:**

- Section 2.1 is missing
- please enlarge the text in the graphs or improve the resolution to allow easy reading in figure 5,6,7,9,10 and 11
- P1 L10,L22 and everywhere: "measures" should be replaced by "measurements" ?
- P4L13 : a word is missing "measurements are available THEY are depicted..."
- P16L9-10 : " The frequency a particular variable is effectively used"

---

## Referee Comment (RC2) · Anonymous Referee #3 · 3 May 2017

**Comments on AMT-2017-30 manuscript**

The work by Hundt et al. "Spectroscopic real-time monitoring of NO2 for city scale modelling", describe a mobile NO2 QCL based spectrometer which has been installed on a tram roof and used for long-term monitoring in the city of Zurich. The data have been compared with data from AQM stations in the city, and a statistical analysis have been performed for concentration maps.

The science in this manuscript has great potential, however the general presentation of the data, the description of the statistical models and the description results has to be improved. I will also suggest authors to look for a better and clearer structure of the paper.

I believe that this work required major revision before being accepted to AMT.

**General comments:**

Further effort should be made on the quality of the writing, the English and the way of presenting data. The paper structure should be reviewed: The main structure: Introduction, methods, results and discussion and conclusions is ok, but the sub-structure should be reviewed. The section 2.4 should be partially included in section 2.3 and 2.5. Section 3.1.should not be split into two sections, same for section 3.3. Finally in figure 10 authors could add the name of the stations so that readers don't need to go to the supplementary material to see station's positions. Finally, authors are making data analysis with 3 seconds averaged data while the instrument response time is larger (9.6 seconds reported in page 3 or 11 seconds in page 9). The whole data analysis should be redone accordingly.

**Specific comments:**

Page 2, Line 8 : « With CLDs, NO2 is determined indirectly as the difference of consecutively measured NOx and NO, making this technique not fast enough for mobile measurements ». To my knowledge CLDs can switch between NO and NOx channel within only few seconds, which I believe is still considered high resolution (<1 hour as mentioned at page 1 line 28). What it can happens is that in case of rapid NOx transitions the value of NO2 can be biased, leading to overestimation or underestimation of NO2 content. Furthermore, since is an indirect approach, it can suffer from artefacts (for instance during the conversion NO2 -> NO in the catalytic oven, other species may also be converted, leading to an overestimation of NO2). I think authors should reformulate this part more in those lines, and please add references.

Page 3, Line 4: « Light passes a 12m … the infrared detector ». Move this below together with the cell description. Could authors comment on the possibility to reduce the cell volume? If my calculation are correct, the height of the cell is 1.8 cm, while usually the beam waist for such a type of propagation is 4-5 mm. It could be possible to reduce the height in order to gain a factor of 2 on the cell volume or there are other factors that will make this volume ideal?

Page 3, Line 17: « We determined the flushing time for a complete gas exchange in the cell to be 9.6 s by exponentially fitting the NO2 signal after switching from outside air to filtered NO2 free air». Is that what expected according to the total volume of the instrument and the gas flow at 100hPa of pressure? Do you see any memory effects that slow down the gas exchange process? Do you observe the same response time while going from outside air to filtered air and on the other-way around?

Page 3, Line 29: « the instrument repeatedly determines the zero-point offset measuring filtered NO 2 -free air ». Mention already here how often this zero-point is determined.

Figure 1a: It could be more interesting to have a picture of the instrument without the cover to be able to see the inside; Figure 1b: Further efforts should be made to provide a clearer and more informative schematic of the setup.

Page 4, Line 13: « The gap between 27 Dec 2015 and 05 Feb 2016 is due to discarded data of reduced quality related to misalignment in the optical system of the instrument. ». It could be interesting to know how many time the instrument required the intervention of the operator over this period of time to show the robustness of the instrument. Any explication on why it took so long (more than 1 month) to realign the optical system?

Page 4, Line 15: « Moreover, data was omitted when the NO2 spectrum was not clearly identified by the processing algorithm ». Authors should specify the percentage of "bad" data that were omitted. Is that related to S/N ratio? Did authors used a threshold on the standard deviation of the fit? More explanations are needed for this point. Replace "data was omitted" with "data were omitted".

Figure 2: « high quality NO2 data ». Authors should be more clear in the manuscript about what good quality data means. Which rules have been used to reject data?

Section 2.3: From making the reading more clear, please specify the different stations (ZUE, SCH, HUE etc) and mention which one are in a background zone and which one in a polluted area.

Page 5, Line 18: « Largest emission sources contributing to ambient NO2 concentrations in Zurich are motorized traffic (47% of NOx) and heating systems (28% of NOx) ». What is the remaining 25%?

Page 6, Line 8: "We used four types:" of what? The object is missing.

Page 8, Line 11 : « Such a correction to the normally calibration free QCLAS method ». Authors should also discuss and if relevant quantifying the effect of missing UV radiation in the inlet of the instrument that would perturb the photo stationary state ratio between NO and NO2. They mention a flushing time of 11sec that is long enough for accounting for an overestimation of the NO2 (due to the reaction of NO with O3 in absence of UV light). A quantification of this artefact should be reported.

Page 8, Line 17: "Discrepancies between the two instruments are mostly due to the higher temporal resolution of the QCLAS and imperfect time synchronization." This should be avoided if QCLAS data are averaged according to the acquisition rate of the CLD (which seems to be the case as mentioned in the caption of figure 4 (1 minute average for both dataset). Were time stamps of the two instruments synchronized before starting the comparison? This statement should not be necessaire if data comparison is done properly.

Page 9, Line 1: "The standard deviation of the QCLAS measurements from the CLD was found to be 0.96 ppb therefore we conclude that the instrument accuracy is about 1 ppb and that the assumption of a constant relative loss of NO2 was correct." Authors should mention the precision of the CLD measurements. This sentence is not clear: is the 0.96 ppb the standard deviation of QCLAS data with respect to CLD data as plotted in figure 4. If so, the sentence "The standard deviation of the QCLAS measurements from the CLD" should be written more as: "The standard deviation of the QCLAS measurements with respect to CLD data". Further discussions about the sensibility of the instrument should arise here. According to the Allan-Werle analysis (Fig.3) the precision of the instrument should be0.02 ppb. Since that is not the case, this means that the AW-deviation should

start to go up in a longer term. I believe that authors should provide an AWW-deviation plot on a longer term to report this long term drifts that will justify this final precision of 1 ppb.

Page 9, Line 10: "We discarded all the positions with corrections exceeding 10 m". Authors should mention the percentage of rejected date due to inconsistent position. Was 10 m decided because of the optimal distance for high resolution data? Was this threshold fixed also according to the GPS position? Discussion on this choice will be appreciated.

Page 9, Line 13: "we added 11.1 seconds to the timestamps of the matched positions in order to account for the flushing time of the measuring cell and for half of the length of the measurement interval." What it should be taken into account is renewing of the sample in the cell, which corresponds to the residence time of the gas in the inlet plus the residence time in the cell. It is not clear here if the 11.1 sec correspond to this. Why half of the length of the measurement interval is also taken into account? At page 3 the response time was estimated to 9.6 sec. Please change the one that is not right.

Page 9, Line 15: "The $NO_2$ measurements refer to 3 seconds averaging time. Therefore, they are strictly speaking not point measurements but refer to a route segment." Not well written. Please rewrite it with a better English. And as mentioned above 3 seconds average is not ideal because the instrument response time is larger.

Page 9, Line 17: "The length of this segment is the product of the integration time of 3 seconds and the speed of the tram. Tram speed associated with the mobile $NO_2$ measurements was smaller than 5 m/s for 64 % and smaller than 10 m/s for 92 % of the time." What is important here is also the average speed of the tram. This will also question the 10 m threshold that authors selected for discharging data because of bad positioning. If the measurement is done in 3 sec and the maximum tram speed is above 10 m/s then the a threshold of 10 m on the position data is not justified and it should be higher in my opinion (It should corresponds to the maximum speed of the tram times the measurement time. eg. 10m/s x 3 s = 30 m threshold).

Page 10, Line 3: "We analyzed the quality of the data obtained in the mobile application twofold: First, we quantified the measurement uncertainty related to noise and variations of the zero point offset. Second, we compared the tram measurements to measurements from a fixed air quality monitoring site (see next section)." What is the mobile application twofold? The whole sentence is not clear and should be rewritten.

Page 10, Line 14: "We corrected the $NO_2$ measurements by linearly interpolating the zero point offsets derived in two consecutive zeroing periods... ... ... Zero point offsets are temporally correlated as the standard deviation of the differences of two consecutive zero point offsets amounts to 2.1 ppb and thus is smaller than 4.3 ppb expected for white noise." I think that this part should be better written. I understand the idea of "predicting" the zero point offset using the two neighbour ones. What is not clear is: is this analysis used to remove "bad" zeroing point? If so, it should be expressed more clearly. Where the 2.1 ppb comes from? From my understanding zero point offsets will be temporally correlated if the difference between the predicted and the measured zero point value (which I will not call standard deviation) will fit within the 4.3 ppb expected in case of white noise. The quantiles are highly reported in this work but not enough used in the discussion.

Page 11, Line 8: Why authors compare measurement from the tram and fix site with different time averages? For fig 6 is clear that measurements with 3 sec integration can show much high variability (due probably to local production). To be more consistent, the comparison should be for the same

time window and same averaging time. Authors should discuss about the higher variability observed at shorter time averaging (3 sec).

Page 11, Line 14: The statement "The calibration of sensors using precise measurements from AQM sites is an option for mobile operated sensors" requires better explanation.

Table 1 Caption: replace number of values with number of measurements. Here too, the quantile values are not well used in the discussion. Authors should either used the more critically or remove them from the table.

Page 11, Line 21: Suggestion to replace "aggregated by 50 meters of track length" with "with 50 meters resolution". Same in the Caption of Figure 8. As mentioned in the caption, the fact that in the figure are the results of the full campaign (and not only 1-way from Bellevue and Bahnofquai should be also mention in the text. Authors should give an idea if how many 1-way trips are used for this plot and how much this corresponds in term of time.

Figure 8. "The tram is directly impacted by traffic between the first and the third stop as well as between the fifth and the seventh stop. There is only little traffic between the third and the fifth stop." Not important in the caption, but this discussion should come out in the text, mentioning that there is then a correlation between the zones which are highly impacted by traffic and the NO2 level measured. What are the "extra" dots in the graph? Out-layer data? This should be specify.

Page 11, Line 21: what authors mean for similar locations? This is something that it should be explained more clearly. What are the criteria for making two locations similar?

Figure 9. There is a miss of general explanation about this analysis. I have the feeling that all for plots at figure 9 don't bring a lot of information from the statistical point of view. I think authors should do a stronger effort in reporting data which are important for the understanding of the work. In the text this figure is not well descripted, ant the readers don't really get the point of what is plotted.
Few more details: The really last red dot at the top right is suspicious… For this measurement period there should be a black dot (MAXTi) on top of the Q095Ti dot.
Modelling period is a bit confusing because to me this is a simple data analysis, and there is not modelling behind. I suggest to change it with measurement period.
The use of "Frequency" on the y-axes to me is also ambiguous, I would suggest to put "number of occurrences". I found weird that in figure c and d the y-axes does not extend to include the whole data set.

Page 16, Line 9: "The frequency a particular variable is effectively used for splitting the 30 minutes data set depends only on the measurements". I guess authors wanted to say that "The frequency **of** a particular variable is effectively used for splitting the 30 minutes data set depend**ing** only on the measurements". Anyway this sentence should be rewritten more clearly.

Page 16, Line 11: The whole paragraph: "The number and spatial distribution of measurements in a modelling period (approximately 500)… …Regression trees based on tram measurements have about 6 location types Ri on average." should entirely be rewritten. I don't see how the number of measurements in the modelling period will limit the possibility of separate different location types?

Section 3.3.1: I am sceptic on meaning of the model version Vt, especially on a such a short time scale. I guess that this approach could be les redundant by increasing the time intervals. For instance using the 30 min intervals to predict other 30 min intervals. Add "respectively" at the end of sentence (line 23).

Page 16, Line 28: "The difference in performance between VAQ and VSTA is small due to the similarity of NO2 concentration encountered at site STA and by the tram-based instrument." Why? VAQ and VSTA are both based on fixed site. How come their similarity should be due to the similarity between NO2 measurement at STA and on the tram?

Table2: How the variables to be used on each model have been chosen? Why there are not variables for VSTA? Same question for section 3.3.2. From what authors describe it seems that variables are chosen to achieve the best overall fit to the measurements at AQM sites. But those variables have a specific meaning and they should not be selected to making the measurement better matching but more according to their physical meaning.

Figure 11 and its discussion: In my opinion, Figure 11 shows that model prediction is less effective at stations far away from the tram track (BLU and HEU). As expected, predictions that use data from the stations (WAQ and WT+A) provide better RMSE but still slope values are more far from unit. I don't think difference between models can be really discussed here. What about error bars in figure 11?

Section 3.4: "Our results suggest that the number of required mobile or static QCLAS instruments for comprehensive NO2 measurement in Zurich is not smaller than the current number of fixed AQM sites." How authors get to this conclusion?

Section 3.4: What about taxis? I guess those could provide a more random and spread-out data set with respect to public transportation which will follow a fix path.

Page 19, Line 19: "The model type the maps are based on is irrelevant in this context." Meaningless.

Page 19, Line 23: "within the same footprint." What does it mean? Same source?

**Technical comments:**

Page 2, Line 10: « Similarly, electrochemical sensors are currently not suited for such applications as their response time is not short enough ». Add a reference

Page 2, Line 25: « a cylindrical multipass cell ». Put the reference Mangold 2016 also here.

Page 2, Line 27: replace 1.1 with 2.1

Page 2, Line 30: remove Additionally

Page 3, Line 14: change "it is not sensitive" with "it is less sensitive".

Page 3, Line 17: « while the air flux is limited by a 100 µm diameter orifice ». Specify to which flow.

Page 3, Line 22: « nano PC ». Specify which one.

Page 3, Line 34: Say what GSM stays for.

Page 4, Line 8: « altering intervals ». I suggest to replace altering intervals with varying intervals.

Page 6, Line 5: replace "in the environment of this location" with "in a specific location".

Page 6, Line 17: replace « for every three seconds » with "every 3 seconds ".

Page 6, Line 21: « First, we focused on small-scale features of the pollutant field. This… …yields a benchmark for the performance of the mobile sensor network » This part should be rewritten more clearly.

Page 7, Line 21: Replace "und" with "and".

Page 7, Line 23: Replace Allan with Allan-Werle everywhere in the manuscript. Is now officially called in this way by the community.

Page 7, Line 30: replace « Such behavior is expected for white noise » with "Such behaviour is expected in case of white noise limited regime".

Page 9, Line 9: replace third with 3.

Page 10, Line 3: Replace "rain" with "bad weather conditions".

Page 10, Line 12: replace "in 20 minutes" with "within a time interval of less than 20 minutes". Replace "omitted" with "rejected".

Page 11, Line 13: Suggestion to replace "is not as good" with "is less pronounced".

Page 12, Line 7: in the sentence "…in 30 minutes modelling periods" add (Ti) which refers then to figure 9.

Page 17, Line 5 and 11: Replace "utilizing" with "using"

Page 17, Line 14: Why authors go back to to VT, VAQ and VSTA described in previous section? The way of presenting things is extremely confusing.

Page 17, Line 19: "The figure shows that the spatial coverage is not optimal, yet, in any version." Why? How authors get to this conclusion?

Arfire et al reference: add the journal.

---

## Author Comment (AC1) · 20 Jul 2017

**Author's response:**

We thank the reviewers for their detailed comments and suggestions. We are confident that we addressed all issues raised by the reviewers appropriately and thus significantly improved the quality of

5 the manuscript. Before discussing all comments in detail, we would like to clarify some important aspects that we treated during the review process.

A main concern for both reviewers was the averaging time of the measured  $NO_2$  concentration. We reported and used 3 s average values for our statistical modeling. The reviewers suggest that this time should be chosen equal to the flushing time of the measurement cell which is about 10 s. However, it

- 10 is important to realize that the flushing of a fully mixed reactor (here the measurement cell) corresponds to a low pass filter. This does not imply that averaging must be done at the response time of this setup. Shorter averaging increases the noise of the individual measurements, while longer averaging leads to loss of information (see e.g. Nyquist criterion). Within that context, we still consider 3 s averages as a good compromise and did not change the time resolution.
- 15 However, after careful reconsideration, we came to the conclusion that adding a time shift to the tram's GPS position corresponding to the instrument response time is not appropriate either. This is because the instrument response is instantaneous (except for a negligible delay caused by the air travelling from the inlet to the measurement cell) even though filtered by a low pass filter. Therefore, we removed that time shift and recalculated the statistical modelling. We exchanged the corresponding
- figures and updated the results tables. This had no significant impact on the results and their interpretation.

Finally, we changed figures 1 and 3 according to the reviewers' comments and added details to the description of the instrument and measurement technique. We also added information about the CLD instruments that were used for comparison and made clear that they are operated in agreement with the

25 European Standard EN14211, which is mandatory for regulatory measurements and assures comparability and quality of the CLD measurements.

In the following we address all comments of both reviewers in detail.

**30 Reviewer #2**

35

**I) General comments:**

**Comment** 1) "The need of mobile measurements is well introduced but an overview of available instruments is missing (commercial or laboratory prototypes). Here CLDs and electrochemical sensors are mentioned and references are given for instruments previously developed by the authors in the MIR. However there is no mention to the many different spectroscopy techniques used in the MIR as well as in the blue spectral region with ECDL or LED. Please compare the performances of the spectrometer developed with these previous works that have to be cited in this paper."

**Response** In the introduction we added further references and included previous works (CAPS, ECDL, LED) as suggested.

Modification "Laser spectroscopy, in contrast to CLD and electrochemical sensors, determines the concentration of NO2 directly and with sampling rates of 1 Hz and higher. Previous studies have shown the applicability of laser diodes in the blue spectral region to measuring NO2 by various cavity enhanced techniques (Courtillot et al., 2006; Fuchs et al., 2009; Kebabian et al., 2005; Osthoff et al., 2006). With higher absorption cross sections than in the visible the mid-infrared region is often chosen for high precision trace gas monitoring of NO2. Here, for instance photoacoustic spectroscopy (Pushkarsky et al., 2006), quartz-enhanced photoacoustic spectroscopy (Patimisco et al., 2014) and direct absorption spectroscopy (McManus et al., 2015) were applied."

**C** 2) "P2,L28" The portable instrument was specifically built for this campaign". Figure 1 is just a scheme that could fit to all multipass spectrometer. The reader expects a detailed description of the technique (iCW mode) and the set-up to understand how the good performances reported in the following are reached. A spectra recorded by the instrument should be shown. Please, detail the "Quantum cascade laser spectrometer" section in this way (see detailed comments in the following)."

**R** We modified the description of the instrument and figure 1 accordingly to make the key elements that contributed to the performance more clear.

20 An absorption spectrum recorded with the instrument was added as figure 1(c).M The modifications can be found with the responses to the detailed comments.

5

10

15

25

30

35

**C** 3) "The statistical model is based on average NO2 measurements during 3s (P6,L18- P9,L15). Note that the response time of the instrument is longer: 9.6s. Averaging at this time scale should be performed prior to analysis."

**R** We reconsidered the data analysis procedure. The instrument's response to a change in the ambient  $NO_2$  concentration is instantaneous except for the negligible travel time of an air parcel from the inlet to the multipath cell. The mixing of air inside the multipath cell effects that the measured  $NO_2$  concentration does not immediately correspond to the  $NO_2$  concentration pumped into the cell. This effect was quantified by these 9.6 seconds. The instrument inherently smooths temporal variations in  $NO_2$  concentration acting as a low pass filter.

We keep the original three seconds sampling rate but leave out the shift of the measurements by 9.6 seconds. Temporal shifting of the measurements is not appropriate. Additional averaging of the data to 9 seconds would further reduce information of the spatial variability of the NO2 concentration field without resolving the smoothing effect inherent to the measurement instrument. We recomputed the statistical models with data processed according to the adapted procedure. Results are reported based on these computations.

Shorter sampling intervals than 3 seconds or alternative sampling schemes (e.g. 1s every 3s) may further improve data quality and the assignment of a NO2 concentration to a position but were not

implemented in this study. We think that 3 second average values are adequate to the chosen complexity of our statistical models.

**C** 4) "NO2 concentration is expected to be sensitive to the solar activity, NO, and O3 concentrations, and to ambient temperature and humidity. Variations of NO2 measurements should be studied according to these different parameters. It is hard to believe that all these parameter effects are removed by the offset correction."

R This is a misunderstanding: Our offset correction does not "remove" any of the mentioned influences on the concentration, but is solely applied to correct for instrumental drifts that appear as a 10 shift in the measured zero. The reviewer is of course correct that ambient NO2 concentrations depend on the mentioned factors. However, the goal here is to determine the ambient NO2 concentration and not to investigate the processes leading to the formation or transformation of NO2. Therefore the task is to minimize sampling artefacts, e.g. through formation of  $NO_2$  in the inlet from reaction of NO with O3, which is inherent for every measurement using a closed measure-15 ment cell. The residence time of the air in the sampling line and the flushing time in the measurement cell have been kept as short as possible, so that the sampling artefacts are minimized. The QCL measurements are corrected to be consistent with the reference CLD measurements. During mobile deployment, the repeated comparison of the QCL measurements and NO2 concentrations measured at the fixed air quality monitoring sites support that the NO2 sampling artefacts (or loss-20 es) are either small or constant on relative terms.

**C** 5) "QCLAS measurements is compared to CLDs measurements taken as reference values. Specification of the CLDs are expected : sensitivity, absolute accuracy, intercomparison of the CLDs..."

**R** CLD is the standard technique for air-quality monitoring of NOx. The instruments are operated according to the European standard EN 14211, including ongoing quality control procedures such as regular calibrations and maintenance by the monitoring networks including intercomparison of the CLDs (Reference: "Technischer Bericht zum Nationalen Beobachtungsnetz für Luftfremdstoffe" (https://www.empa.ch/documents/56101/246436/Technischer+Bericht+2016/0bc321a3-f489-4f20-bcda-a323fbc4ca8a), 2016).

We added detailed specifications of the performance of the CLDs as determined within the Swiss
 Air Pollution Monitoring Network NABEL.

**M** "The CLD has a limit of detection of 0.1 ppb and at 30 ppb a total uncertainty of 1.25 ppb (NABEL, 2016)."

**35 **Detailed comments**

5

25

**C** "**a**) In the abstract and the introduction: please mentioned the technique used and the wavenumber of the laser."

**R** Done as suggested.

**M** "The NO2 concentration is measured by direct absorption spectroscopy at 1600 cm-1."

**C** "**b**) Introduction:

5

15

20

25

1) This work is dedicated to urban monitoring, please specify the NO2 concentration range to clarify the instrument performances required in term of detection limit.

2) Insert references to previous NO2 spectrometers (see General comments)

3) Please gives accurate numbers concerning instrument performances:

P1, L30 "accurate"?

P2, L9: "not fast enough"?

10 P2 L11 "not short enough" P2 L13 "relatively high"

P2 L16: "high sampling rates"

**R** The comments above were implemented as suggested.

C "c) 1.1 QCL spectrometer :"

Ten comments concerning the QCL spectrometer were made. In the following we will address these comments individually.

**C** "1) The iCW operation was published in Fischer 2014, reporting on a different instrument that operates at 7.7µm. In this paper, a short description of this technique is expected and a detailed description of the optical set-up at 6.25µm as well. I could not find mention to an optical etalon : How is obtained the frequency linearization ?"

**R** We thank the reviewer for pointing this out. We modified the respective sections to include details about icw driving, the optical set-up and the frequency linearization with a 2" Ge etalon.

**M** "In iCW mode the drive current of the QCL is dropped to zero between individual pulses leading to rapid heating and therefore frequency tuning during each pulse. The pulses are generated by discharging a capacitor over the QCL. A frequency tuning of >1 cm-1 was achieved (see Figure 1(c). By shaping the current ramp with different RC-elements, close to linear tuning could be achieved. See (Fischer et al., 2014) for details.

...For frequency calibration a 5.08 cm Ge-etalon can be inserted between the laser and the first mirror."

30

35

**C** "2) Please show a recorded spectra around 1600 cm-1. How many NO2 lines do you monitor ? Is the spectra large enough to monitor water ?"

**R** A typical experimental spectrum of  $NO_2$  and water absorption lines is included (Figure 1(c)). In fact, the water absorption line was always fitted along the  $NO_2$  and used as a frequency reference for the  $NO_2$  absorption and as a criterion to determine if the spectrum is fitted correctly.

**M** A spectrum was added as Figure 1(c).

**C** "3) In figure 1 : the detailed optical scheme would be more useful than the schematic content of the instrument"

**R/M** We replaced figures 1a and 1b with a detailed optical scheme and a photo of the instrument without cover (also following the suggestion of Reviewer #3).

**5**

**C** "4) P3, L17 : please specify that the 100 $\mu$ m orifice is placed at the inlet of the cellule and give the corresponding flow value."

**R** Done as suggested.

**Μ** "...while the air flux is limited to 180 sccm by a 100 μm diameter orifice placed at the gas inlet."

10

**C** "5)P3 L17-19 : The response time is a main concern. It is measured to be equal to 9.6s by switching from NO2 free air to NO2 sample. Is this really consistent with the complete exchange time of the cell according to the flow and pressure values?"

R The flushing time can also be estimated from the flow through the orifice which is about
 15 180 sccm (=3.04 mbar·l/s). With a volume of 0.3 I and a pressure of 100 mbar the resulting flushing time would be about 10 s which is in good agreement with the measurement.

C "6) P3L22 : what is the acquisition rate ?"

**R** Spectra are generated at a rate of 2 kHz. However, due to the data transfer limitations of the DAQ system, only 50 % duty-cycle acquisition is achieved, i.e. 1000 spectra are averaged within 1 s. We will make this clear in the text.

**M** "After acquisition of 1000 spectra at a rate of 2 kHz the data is transferred via a USB 2.0 port to a nano PC (Zero pro, Xi3 corporation) where the spectra are averaged and analyzed by a custom-written LabVIEW program."

25

20

**C** "7)P3, L23-25: Concerning the fitting procedure : Please indicates which parameters are fixed (such as P probably) or left variable. The instrument not being temperature stabilized, what about the temperature parameter ? Is the path length deduced fixed?"

R We agree that this should be mentioned. The pressure in the cell is constant and could therefore
 be fixed. The temperature of the cell is measured by a thermistor and the actual temperature is fed
 back to the fitting routine. As the cell is a monolithic ring the path length is constant and therefore
 fixed in the fitting routine as well.

**M** "Pressure and path length are considered constant, while the temperature of the cell is measured with a thermistor and fed back to the fitting program."

35

**C** "8) Water absorption has strong absorption lines around 1600cm-1, and its concentration is changing during tram measurements. How is that taken into account ? Do you include a fit of water line to monitor it ? is it included in the base line?"

**R** Indeed, water has strong absorptions near our target NO2 line. The pressure broadened wing of this line is included in the baseline of the fit therefore changes in water concentration do not change the result of the fit. Furthermore, a small HDO line is fitted along the NO2 to monitor the water concentration and to serve as a frequency reference when the NO2 concentration is close to our detection limit. The fitted lines can be seen in the spectrum that was added in response to *General comment 2*.

C "9) P3,L29: please make explicit what you call "zero-point offset"."

5

15

30

R We will rephrase to make clear what is meant by "zero point offset".

10 **M** "Second, the instrument repeatedly determines the zero point offset, i.e. the instrument signal at zero  $NO_2$ , by measuring filtered  $NO_2$ -free air for 2 minutes every 20 minutes."

C "10) The dynamic range of the instrument should be given somewhere in the paper"

**R** We agree and added information about the dynamic range of the instrument. The dynamic range goes from from 1 ppb to several ppm in concentration.

**M** "The chosen  $NO_2$  absorption feature at 1599.9 cm-1 offers a large dynamic range from 0 to several ppm which is suitable for ambient concentration measurements in cities."

C "d)2.3. NO2 concentration measurements at fixed sites

20 Indicates that it is performed by CLD. Please give the specifications of these instruments (see general comments)."

**R/M** We will make clear that the instruments at the fixed sites are CLDs and give their specifications.

25 C "e) P5 L17 : " NO2 concentrations encountered in Zurich are moderate compared to other cities."
 Please specify here the typical range."

**R** We added information about the annual mean concentrations to the manuscript. Additionally, figure 6 shows a 24 h time series of NO2 concentrations.

**M** "The reported annual mean NO2 concentrations in 2015 for the AQM sites were: HEU: 18  $\mu$ g/m3, ZUE: 31  $\mu$ g/m3, STA: 33  $\mu$ g/m3, SCH: 45  $\mu$ g/m3, SWD: 47  $\mu$ g/m3, RGS: 50  $\mu$ g/m3 (OSTLUFT, 2015)."

C "f) 2.5 Statistical modelling

1)P6, L18 : NO2 concentrations should be averaged to 9.6s, the response time (see general comments)"

35 **R** See our response to general comment 3).

**C** "2)Please provide the reference (Breinam1984) concerning the "regression tree approach at first mention: P6, L32"

**R** We provided the reference as suggested.

**5 **C** "g) 3.1. Instrument performance and stability"**

Comments concerning the Instrument performance and stability were made. In the following we will address these comments individually.

**C** "1) The Allan deviation is expected to increase at long integration time due to technical drifts or ambient parameters change. Can the author provide a longer set of measurements to show when it appears ?

10 it appears ?

30

**R** We agree. The Allan-Werle plot (Figure 3) was updated using a longer time series. Now the onset of the drift after an integration time of about 400 seconds can be observed.

C "2) In figure 3, in the upper panel can you confirm that Dark-Noise measurements are the upper
 one or is there a color inversion ? I would understand an offset of about 1.5ppb is the
 measured in" NO2 free air" rather than Dark-Noise, from a non perfect filtering or other
 effects."

**R** There is no color inversion. Both traces should be around zero. It is rather an offset that was added to the zero-air measurement (red trace) for clarity.

20 M "The dark-noise trace is offset by 1.5 ppb from zero for clarity."

**C** "3)P8, L1 : Concerning the "measurement strategy" I understand how the drift is corrected but this process relies on the zero NO2 gas production in-situ. Did you test that there is no NO2 trace in this gas ? I would expect some NO2 trace at the ppb level (according to fig 1).

25 And the residual concentration is expected to depend on NO2 concentration in ambient air. Furthermore, the sensitivity to water concentration should be investigated . Can you provide some details ?"

**R** Indeed, our drift correction fully relies on the quality of the in-situ zero-air generation. The Purafill chemisorbant is specified to have a minimal initial removal efficiency of 99.5%. That would leave a maximal trace of 0.1 ppb at the average ambient concentration in Zurich, which is 20 ppb. For a trace of 1 ppb an ambient concentration of 200 ppb would be necessary. The chemisorbant is specified to perform from 10-95% relative humidity and between -20 and 51°C. Laboratory tests confirmed that there is no detectable NO2 after the filter.

C 3b"Another aspect should be detailed : how is calibrated the correction, on which reference measurement ? Did you performed a referenced measurement prior in-situ measurements ?
 Did you repeat it during the measurement campaign?"

**R** The linear correction to the NO2 concentration values obtained by the QCLAS instrument was found by comparison with a calibrated CLD from the fixed air quality monitoring station in Dübendorf. The instrument measured in parallel with the CLD for several days. Figure 4 shows the agreement of the CLD and QCLAS measurements (slope 1) after the correction was applied. Such a calibration was not repeated during the measurement campaign. However, as detailed in the paper, we compared the QCLAS measurements in the vicinity of air-quality monitoring stations to the values reported by the stations (Figures 7a and b). The comparison showed very good agreement (slope 1.03) at the site STA, where a relatively small temporal and spatial variability is expected. Based on these measurements we are confident that the calibration did not change during the measurement campaign.

**C** "4) Allan deviation only demonstrate the stability of the instrument but not the accuracy. Caption of figure 3 is to be modified and 2 P7, L24-25 : "precision" should be replaced by "stability""

R Here must be a misunderstanding, because on P7, L24-25 we stated: "The best precision of
 30 ppt is reached after 100 s averaging while the 1 s precision is about 300 ppt." and in the caption of figure 3: "Determination of the instrument precision and stability: Time series of zero-air (black) and detector dark-noise (red) and the associated Allan-deviation plots." There is no mention of "accuracy" here.

In addition, we do not agree that "precision should be replaced with stability" as the integration time of the Allan plot (x-axis) is a measure for stability, while the best precision can be seen from the corresponding Allan-deviation (y-axis).

**C** "5) P8, L8-14 : I would expect the correction factor to change in time due to passivation of the surfaces or water interferences : the calibration performed in June is probably different

25 during winter. Did you test this point ?

5

10

20

30

**R** We agree that the correctness of the calibration factor has to be verified. For this reason we compared the QCLAS measurements to measurements at fixed air-quality monitoring sites when the instrument was in their vicinity which yielded good agreement (see also response to *comment 3b*). Therefore we are confident that the calibration factor did not change significantly. Water interferences are excluded by the line selection as detailed in response to *comment c*) 8).

C "6)The correction factor of 1.2 is applied. But how was the instrument firstly calibrated?"R Please see response to comment g)3b).

35 **C** "7)P9L2: Please specify the absolute accuracy of the CLD to justify the 1ppb accuracy of the instrument."

**R** The CLD has a total uncertainty of 1.25 ppb at 30 ppb concentration (Reference: "Technischer Bericht zum Nationalen Beobachtungsnetz für Luftfremdstoffe"

(https://www.empa.ch/documents/56101/246436/Technischer+Bericht+2016/0bc321a3-f489-4f20-bcda-a323fbc4ca8a), 2016). The accuracy of the QCLAS instrument has to be considered relative to the CLD which is the standard instrumentation for ambient air quality monitoring (see European Norm EN 14211). This will be changed in the manuscript to make it clear.

5 **M** "The CLD has a limit of detection of 0.1 ppb and at 30 ppb a total uncertainty of 1.25 ppb (NABEL, 2016)."

"...therefore we conclude that the instrument accuracy is about 1 ppb with respect to the CLD and that the assumption of a constant or negligible relative loss of NO2 was correct."

10 **C** "8) P9L15-19 : as mentioned previously, measurements should be averaged to 9.6s resulting in a longer route segment."

**R** For the reasons stated above we are convinced that using 3 second averages is appropriate.

C "h) 3.2 Intra-urban and temporal variation in NO 2 concentration"

15 Five comments concerning Intra-urban and temporal variation in NO2 concentration were made. In the following we will address these comments individually.

**C** "1) Fig 6 : in the caption please clarify which AQM station (STA or SCH) is close to which terminal stations (Seebach or Triemli)"

**R** We indicate the passing of the tram at the terminal stations primarily to show that the tram is in 20 regular operation on October 23, 2015. A direct comparison of QCLAS measurements and measurements from the AQM sites is presented in Figure 7. Travel time of the tram service number 14: Seebach  $\rightarrow$  AQM STA (+ ~16 min)  $\rightarrow$  AQM SCH (+ ~16 min)  $\rightarrow$  Triemli (+ ~7 min) [total: ~39 min].

C "2)P11L7 : again measurements should be averaged to 9.6s."

25 **R** See our answer above.

30

**C** "3)P11,L15-17: I don't understand why "this analysis clearly shows the advantage of a mobile measurement device that does not rely on fixed sites for the determination of its calibration parameters". I would assume that the main reason for the discrepancies observed with SCH measurements is due to the difference of averaging time : 9.6s response time for the QCLAS and 1min for the CLD. SCH being impacted by heavier traffic than STA , fast events of strong pollution will have more impact."

**R** The wording was not clear enough and has been improved.

M "This can be expected for two reasons: (i) the larger distance, (ii) the location of site SCH at a
 crossroad where traffic flow is controlled by traffic lights and where the short term variability of NO2
 is larger. From a more general perspective, this is of importance when mobile sensor data is corrected
 based on data from nearby fixed reference sites, as suggested by (Arfire et al., 2015; Saukh et al.,

2015). The parameters for data correction as determined from such an approach depend on the local concentration variability and the time response of the sensor and the reference instrument."

C "4) Figure 8 : What is the meaning of points after the 7th stop ?"

- 5 **R** The route section shown in Figure 8 is not congruent with the tram stops, i.e. the route section is longer than the distance given by the tram stops and its length is not an exact multiple of 50 m. We suppress plotting of any boxplots for which the corresponding number of measurements is below 20. This was the case for the boxplot on the right side of the Figure to which the reviewer referred.
- 10 **M** The figure has been updated accordingly.

**C** "5) In this section, temporal variation of NO2 should be studied as a function of other parameters : NO, O3, H2O,T,solar activity (see general comments)."

R We thank the reviewer for the remark, however, the suggested study is beyond the scope of our manuscript. Here we mainly focus on the development, applicability and validation of a compact and mobile laser spectrometer for ambient NO2 measurements. As mentioned earlier, this work does not investigate the processes leading to the formation or transformation of NO2 and influencing the spatial variability, but rather provide a concept for spatial mapping of urban NO2. The statistical models directly rely on measurements that refer to confined 30 minutes time periods. The NO2 concentration at any location is assumed to be constant within 30 minutes. Temporal variations in concentration due to changes in emissions and meteorology are captured by variations in the measurements of consecutive 30 minute time periods. The statistical models are not intended to offer a full atmospheric chemistry and transport analysis but to map the instantaneous NO2 concentration field.

25

**C** "**i)** Model validation by means of three seconds NO 2 measurements from the tram Should be performed with 9.6s averaging measurements"

**R** See our reply above.

30 **C** "j) 3.3.2 Model validation by means of 30 minutes mean NO2 concentrations at fixed AQM stations

1) Fig 10: change the color code of NO2 concentration to allow to distinguish between very low concentrations (presently in white) from the zone where the model cannot compute concentration. Presently, the spatial coverage is not clear."

35 **R** The figure was replaced. Areas for which no NO2 concentrations are predicted based on the statistical models are now depicted in white and denoted as "NA". The concentrations of the model predictions are well above zero and therefore depicted in yellow (and red). **C** "2) P17, L29-30 : " The obtained RMSE values range between 1.5 ppb at HEU (elevated background site) and 7.5 ppb at RGS (roadside)." Given values are not consistent with Fig 11."

**R** These numbers do not refer to Figure 11 but to an additional analysis that quantifies the magnitude of the accuracy of the  $NO_2$  concentration of a location type estimated by statistical models based on measurements from one tram instrument.

In this analysis we quantified the error that may result when the 25 minute mean concentration is estimated based on the complementary measurements (5 minutes) of a 30 minutes interval (6 alterations). This error is 1.5 ppb for HEU and 7.5 ppb for RGS.

- The elevated AQM site HEU is not directly impacted by traffic and encountered an annual mean
   NO2 concentration of 18 ug/m3 in 2015. Accordingly, temporal NO2 concentration variations are small. In contrast, AQM site RGS is heavily impacted by traffic resulting in an annual mean NO2 concentration of 504 ug/m3 in 2015. Temporal variations in NO2 concentration are much higher. We rephrased this paragraph in order to better present our findings.
- M "In our statistical models, the mean of all the measurements in Ri (covering approximately 1/m of 25 minutes with mostly m = 6) is taken as the estimate value for the 30 minute mean concentration of location type Ri. We performed the following computation in order to analyze the agreement between the mean derived from a limited number of measurements and the true 30 minute mean concentration for several locations. For this analysis, measurements from single AQM sites from the year 2015 were used. We computed 5 minute means and complementary 25 minute means (6 pairs in 30 minutes, respectively). The obtained RMSE values range between 1.5 ppb at HEU (elevated back-ground site) and 7.5 ppb at RGS (roadside). Therefore, differences between the mean of a subset of tram measurements referring to a particular location type and its 30 minute mean have to be expected if NO2 concentration at this location is highly variable on the short-term."

25

5

**C** "3) What is the error/shift value that can be attributed to the absolute calibration of the different CLDs ? Were they regularly calibrated on the same gas sample ?"

**R** The total uncertainty of 1.25 ppb at 30 ppb concentration was previously discussed. The CLDs of the AQM sites are regularly (25 h intervals) checked and calibrated.

30

**C "k) Conclusion**

1)P20L2 : " QCLAS spectrometer that operated autonomously.... over a period of four months" should be added a mention to the intervention for optical alignment

**R** Changed as suggested.

35

**C** "2) Please indicate clearly that the 2 ppb agreement is obtained after a correction factor of 1.2 derived from these measurements "

R Done.

**C** "3) 3s to be replaced by 9.6s."

**R** For the reasons stated above we are convinced that using 3 second averages is appropriate.

**5 **C** "Typesetting:**

- Section 2.1 is missing
- please enlarge the text in the graphs or improve the resolution to allow easy reading in figure

5,6 ,7,9,10 and 11

- R The original figures that were copied into the MS-Word template are in PNG or PDF-Format. We
   expect less compression and better quality when the figures are integrated during final type-setting.
  - P1 L10,L22 and everywhere: "measures" should be replaced by "measurements" ?
  - P4L13 : a word is missing "measurements are available THEY are depicted..."
  - P16L9-10 : " The frequency a particular variable is effectively used" "
- 15 **R** We implemented the necessary changes.

20 Reviewer #3

25

35

**General comments:**

**C** Further effort should be made on the quality of the writing, the English and the way of presenting data. The paper structure should be reviewed: The main structure: Introduction, methods, results and discussion and conclusions is ok, but the sub-structure should be reviewed. The section 2.4 should be partially included in section 2.3 and 2.5. Section 3.1.should not be split into two sections, same for section 3.3. Finally in figure 10 authors could add the name of the stations so that readers don't need to go to the supplementary material to see station's positions. Finally, authors are making data analysis with 3 seconds averaged data while the instrument response time is larger (9.6 seconds reported in page 3 or 11 seconds in page 9). The whole data analysis should be redone accordingly.

30 done accordingly.

**R** We appreciate the general comments. We are convinced that using sections and subsections increases the readability and therefore prefer to keep the structure of the manuscript as is.

Figure 10 is updated also following the comment of Reviewer #2.

The flushing time of the instrument and the temporal resolution used in the data analysis are addressed in detail in response to reviewer #2's comments. In short: Sampling at a higher frequency than the flushing time is necessary. But we were mistaken to add the flushing time as a delay to the instrument response and therefore redid the data-analysis and statistical modelling without that delay.

**Specific comments**

- C Page 2, Line 8 : « With CLDs, NO2 is determined indirectly as the difference of consecutively measured NOx and NO, making this technique not fast enough for mobile measurements ». To my knowledge CLDs can switch between NO and NOx channel within only few seconds, which I believe is still considered high resolution (<1 hour as mentioned at page 1 line 28). What it can happens is that in case of rapid NOx transitions the value of NO2 can be biased, leading to overes-timation or underestimation of NO2 content. Furthermore, since is an indirect approach, it can suffer from artefacts (for instance during the conversion NO2 -> NO in the catalytic oven, other species may also be converted, leading to an overestimation of NO2). I think authors should reformulate this part more in those lines, and please add references.
- R The reviewer is correct that CLDs can switch between NO and NOx channels within a few seconds, however, instruments typically use a low pass data filtering algorithm leading to a much slower time response. For example, the response time of the CLD used within the Swiss Air Pollution Monitoring Network (Horiba APNA-360) is 120 seconds which is not fast enough for mobile measurements (for example on a tram). "High resolution" as mentioned in page 1 line 28 refers to the temporal resolution of a pollution map.
- 20 We agree with the reviewer concerning the sources of uncertainty of CLD NOx measurements and will modify the section accordingly.

**M** "The indirect measurement approach of the CLD instruments can lead to biased  $NO_2$  measurements when NO and  $NO_2$  are temporally highly variable. Furthermore, the conversion of  $NO_2$  to NO in a catalytic converter can lead to measurement errors as other species such as  $NH_3$ ,  $HNO_3$  and  $HNO_2$  can be converted, too, and therefore lead to  $NO_2$  signals that are biased high (NABEL, 2016; Steinbacher et al. 2007)."

**C** Page 3, Line 4: « Light passes a 12m ... the infrared detector ». Move this below together with the cell description. Could authors comment on the possibility to reduce the cell volume? If my calculation are correct, the height of the cell is 1.8 cm, while usually the beam waist for such a type of propagation is 4-5 mm. It could be possible to reduce the height in order to gain a factor of 2 on the cell volume or there are other factors that will make this volume ideal?

**R** We moved the cited sentence as suggested. In principle, the reviewer is right: If all reflections were in the same plane the cell volume could be reduced by half the volume. However, the off-axis configuration was found to be favorable in terms of optical noise (fringes). As such, the reflections alternate between two horizontal levels on the mirror, and thus, there is very little room for further reducing of the cell volume.

35

30

**C** Page 3, Line 17: « We determined the flushing time for a complete gas exchange in the cell to be 9.6 s by exponentially fitting the NO2 signal after switching from outside air to filtered NO2 free air». Is that what expected according to the total volume of the instrument and the gas flow at 100hPa of pressure? Do you see any memory effects that slow down the gas exchange process?

Do you observe the same response time while going from outside air to filtered air and on the other-way around?

**R** We detailed the gas flow and exchange time in a response to *comment c)4*) of reviewer #2. Yes, the response time is the same in both directions.

C Page 3, Line 29: « the instrument repeatedly determines the zero-point offset measuring filtered NO2 -free air ». Mention already here how often this zero-point is determined.
 R Done.

M "...by measuring filtered NO2-free air for 2 minutes every 20 minutes."

5

15 **C** Figure 1a: It could be more interesting to have a picture of the instrument without the cover to be able to see the inside; Figure 1b: Further efforts should be made to provide a clearer and more informative schematic of the setup.

**R** We agree and exchanged the figures also in accordance with comments of reviewer #2.

- **C** Page 4, Line 13: « The gap between 27 Dec 2015 and 05 Feb 2016 is due to discarded data of reduced quality related to misalignment in the optical system of the instrument.». It could be interesting to know how many time the instrument required the intervention of the operator over this period of time to show the robustness of the instrument. Any explication on why it took so long (more than 1 month) to realign the optical system?
- **R** From 27 Dec 2015 to 05 Feb 2016 there was only one intervention by the operator: The instrument was removed from the tram and taken to the laboratory on 27th January 2016 and installed again on the tram on 4th February 2016 after realignment and testing. It did not take one month to realign the instrument, but the data from 27th December to after the repair had to be discarded.
- 30 **C** Page 4, Line 15: « Moreover, data was omitted when the NO2 spectrum was not clearly identified by the processing algorithm ». Authors should specify the percentage of "bad" data that were omitted. Is that related to S/N ratio? Did authors used a threshold on the standard deviation of the fit? More explanations are needed for this point. Replace "data was omitted" with "data were omitted".
- R In total 37 % of the measurement days yielded data that could be used for statistical modeling.
   We used the water concentration derived from the measured spectrum to flag measurement periods where the fitting algorithm did not work correctly. The according information is added in the manuscript.

**M** "The water concentration determined from the measured spectrum was used as an indicator to flag such data. In total 37 % of the measurement days yielded data that could be used for statistical modeling."

5 **C** Figure 2: « high quality NO2 data ». Authors should be more clear in the manuscript about what good quality data means. Which rules have been used to reject data?

**R** The rules for rejection of data are stated in section 2.2.

Data were used if...

- ...the tram is not in the depot
- ...its position can be attributed to the track
- ...the NO2 spectrum is clearly identified (which is the case when the water concentration measured alongside the NO2 is between 0.1 and 4%)

C Section 2.3: From making the reading more clear, please specify the different stations (ZUE, SCH,
 HUE etc) and mention which one are in a background zone and which one in a polluted area.
 R In response to *comment e*) of Reviewer #2 we added information about the annual mean NO2

concentrations at the different sites to section 2.3

**M** The reported annual mean NO2 concentrations in 2015 for the AQM sites were: HEU: 18  $\mu$ g/m3, ZUE: 31  $\mu$ g/m3, STA: 33  $\mu$ g/m3, SCH: 45  $\mu$ g/m3, SWD: 47  $\mu$ g/m3, RGS: 50  $\mu$ g/m3.

20

25

10

**C** Page 5, Line 18: « Largest emission sources contributing to ambient NO2 concentrations in Zurich are motorized traffic (47% of NOx) and heating systems (28% of NOx) ». What is the remaining 25%?

**R** The missing 25% are: 15% industry, 9% constructions and 1% agriculture and forestry ("Luftbilanz der Stadt Zürich", 2012). We added the information to the manuscript.

**M** Largest emission sources contributing to ambient  $NO_2$  concentrations in Zurich are motorized traffic (47% of  $NO_x$ ) and heating systems (28% of  $NO_x$ ) (followed by industry (15%), constructions (9%) and others (1%)) (Brunner and Scheller, 2014). There is no heavy industry.

30 C Page 6, Line 8: "We used four types:" of what? The object is missing.
R The sentence should be: "We used four types of traffic intensities:"
M ...of traffic intensities:

C Page 8, Line 11 : « Such a correction to the normally calibration free QCLAS method ». Authors
 should also discuss and if relevant quantifying the effect of missing UV radiation in the inlet of the instrument that would perturb the photo stationary state ratio between NO and NO2. They mention a flushing time of 11sec that is long enough for accounting for an overestimation of the NO2

(due to the reaction of NO with O3 in absence of UV light). A quantification of this artefact should be reported.

**R** As mentioned in our response to reviewer #2, sampling artefacts, e.g. through formation of  $NO_2$ in the inlet from reaction of NO with  $O_3$  is inherent for every measurement using a closed measurement cell. The residence time of the air in the sampling line and the flushing time in the measurement cell have been kept as short as possible, so that the sampling artefacts are minimized. The residence time for the reference CLD instruments is comparable (about 6s in sampling line plus mixing time in measurement cell), so it can be expected that the sampling artifacts of the reference  $NO_2$  measurements (which are performed in agreement with the European Norm EN 14211 which is mandatory for regulatory measurements) are comparable to the QCLAS measurements.

A quantification of this measurement artefact is not possible because the mobile measurement platform was not equipped with NO and  $O_3$  measurements. However, a back of the envelop calculation suggests that the artefact is below 1 ppb (assuming the rate constant for the reaction of NO and  $O_3$  at 298 K being 0.0004 1/(ppb·s) and NO and  $O_3$  concentrations of 20 ppb and 10 ppb, respectively – note that the measurements have been done during the cold season at low  $O_3$  concentrations).

**M** We added the following sentences: "Note that formation of  $NO_2$  in the sampling line of instruments due to reaction of NO with  $O_3$  can lead to an overestimation of  $NO_2$ . The influence of this artefact is small, especially because the measurements have been done during the cold season at low O3 concentrations. It is, however, for the QCLAS a bit higher than for the reference CLD instrument, because the residence time of the air sample is for the CLD somewhat shorter (about 6s)."

**C** Page 8, Line 17: "Discrepancies between the two instruments are mostly due to the higher temporal resolution of the QCLAS and imperfect time synchronization." This should be avoided if QCLAS data are averaged according to the acquisition rate of the CLD (which seems to be the case as mentioned in the caption of figure 4 (1 minute average for both dataset). Were time stamps of the two instruments synchronized before starting the comparison? This statement should not be necessaire if data comparison is done properly.

R We agree that the above statement might be a bit confusing. We are comparing 1 minute average values of a CLD and the laser spectrometer. The CLD has a response time of 60 seconds and the laser spectrometer has 9.6 seconds flushing time. Single pollution events of less than 60 seconds duration cannot be captured in the same way and therefore lead to different concentration signals. The time synchronization was done properly and therefore imperfect time synchronization cannot explain discrepancies.

35 **M** Text has been changed accordingly.

5

10

15

20

25

**C** Page 9, Line 1: "The standard deviation of the QCLAS measurements from the CLD was found to be 0.96 ppb therefore we conclude that the instrument accuracy is about 1 ppb and that the assumption of a constant relative loss of NO2 was correct." Authors should mention the precision of the CLD measurements. This sentence is not clear: is the 0.96 ppb the standard deviation of QCLAS data with respect to CLD data as plotted in figure 4. If so, the sentence "The standard deviation of the QCLAS measurements from the CLD" should be written more as: "The standard deviation of the QCLAS measurements with respect to CLD data". Further discussions about the sensibility of the instrument should arise here. According to the Allan-Werle analysis (Fig.3) the precision of the instrument should be0.02 ppb. Since that is not the case, this means that the AW-deviation should start to go up in a longer term. I believe that authors should provide an AWW-deviation plot on a longer term to report this long term drifts that will justify this final precision of 1 ppb.

10 **R** In response to *General comment 5*) of Reviewer #2 we stated the precision of the CLD that was used for the comparison and we will rephrase as suggested.

The Allan-Werle plot (Fig. 3) was updated with a larger data set to show the drift of the instrument after about 400 seconds.

1 ppb is the accuracy of the laser spectrometer in comparison with the CLD of the AQM site after calibration and zero point correction.

**M** Done as suggested.

**C** Page 9, Line 10: "We discarded all the positions with corrections exceeding 10 m". Authors should mention the percentage of rejected date due to inconsistent position. Was 10 m decided because of the optimal distance for high resolution data? Was this threshold fixed also according to the GPS position? Discussion on this choice will be appreciated.

**R** The accuracy of GPS measurements in densely built environments can be deteriorated due to the reception of signals from fewer satellites than maximally possible and due to multipath effects in GPS signal propagation. The 10 m threshold is arbitrary but this approach is straightforward. GPS position accuracy can be expected < 5 m under good conditions. The tram is bound to the tracks. The larger the distance between GPS position and the nearest tram track the less reliable is the GPS position. The reliable assignment of the NO2 measurement to a position is required so that the data can be used in the statistical models.

**M** We suggest to not to extend the corresponding section.

30

35

5

15

20

25

**C** Page 9, Line 13: "we added 11.1 seconds to the timestamps of the matched positions in order to account for the flushing time of the measuring cell and for half of the length of the measurement interval." What it should be taken into account is renewing of the sample in the cell, which corresponds to the residence time of the gas in the inlet plus the residence time in the cell. It is not clear here if the 11.1 sec correspond to this. Why half of the length of the measurement interval is also taken into account? At page 3 the response time was estimated to 9.6 sec. Please change the one that is not right.

**R** In the data analysis of the initially submitted version of the manuscript we subtracted a time constant of 11.1 seconds from the time of the  $NO_2$  measurement and related the  $NO_2$  measurement to the position at this time. The time constant was the sum of 1.5 seconds (the half of the 3 second averaging period) and 9.6 seconds (time constant for  $NO_2$  changes in the measurement cell). We changed the data analysis in the revised manuscript by omitting the 9.6 seconds.

**M** "Hereby, we added 1.5 seconds to the timestamps of the matched positions. Such a delay is necessary because at a given time-stamp the GPS provided the present position while the  $NO_2$  concentration is the average of 3 seconds before a time-stamp."

**C** Page 9, Line 15: "The NO2 measurements refer to 3 seconds averaging time. Therefore, they are strictly speaking not point measurements but refer to a route segment." Not well written. Please rewrite it with a better English. And as mentioned above 3 seconds average is not ideal because the instrument response time is larger.

**R Done.**

**M** "The NO2 measurements represent 3s averaged data, which, considering the traveling speed of the tram, cover a given route segment instead of representing a fixed point measurement."

15

20

25

30

35

10

5

**C** Page 9, Line 17: "The length of this segment is the product of the integration time of 3 seconds and the speed of the tram. Tram speed associated with the mobile NO2 measurements was smaller than 5 m/s for 64 % and smaller than 10 m/s for 92 % of the time." What is important here is also the average speed of the tram. This will also question the 10 m threshold that authors selected for discharging data because of bad positioning. If the measurement is done in 3 sec and the maximum tram speed is above 10 m/s then the a threshold of 10 m on the position data is not justified and it should be higher in my opinion (It should corresponds to the maximum speed of the tram times the measurement time. eg.  $10m/s \times 3 s = 30$  m threshold).

**R** The distance between tram stations in Zurich is normally a few 100 meters. The actual tram speed depends on track characteristics (e.g. curves, traffic lights) and maximum acceleration rates. At each station the tram stops for the entry/exit of passengers. Additional stops may occur at traffic lights.

The GPS receiver on the tram operates independently from the QCLAS instrument. First, outliers in the GPS positions have to be identified. If a GPS position is located more than a distance D away from the tram track the true position error is equal or larger than D. 10 m corresponds to about three times the accuracy of a single GPS position. Second, the GPS positions were orthogonally projected on the tram tracks as the tram is bound to the tracks. Third, the position of the NO2 measurement is interpolated from the improved GPS positions.

This means:

- Continuous measurement of NO2 with QCLAS in (ti,ti+3s) → Average NO2 concentration for (ti+1.5s)
- 2. Improved GPS position at tk, tk+3s, tk+6s, ...
- 3. Interpolation of tram position at ti+1.5s

Obviously, the error of a single GPS position impacts the resulting position of the  $NO_2$  measurement and also the link between the  $NO_2$  measurement and the spatial information used in the statistical models.

**M** We suggest keeping the text as it is.

5

**C** Page 10, Line 3: "We analyzed the quality of the data obtained in the mobile application twofold: First, we quantified the measurement uncertainty related to noise and variations of the zero point offset. Second, we compared the tram measurements to measurements from a fixed air quality monitoring site (see next section)." What is the mobile application twofold? The whole sentence is not clear and should be rewritten.

10

30

**R** Twofold means: "In two ways" or "twice". We modified the text accordingly to avoid confusion.**M** "We analyzed the quality of the data obtained in the mobile application in two ways."

C Page 10, Line 14: "We corrected the NO2 measurements by linearly interpolating the zero point offsets derived in two consecutive zeroing periods... ... Zero point offsets are temporally correlated as the standard deviation of the differences of two consecutive zero point offsets amounts to 2.1 ppb and thus is smaller than 4.3 ppb expected for white noise." I think that this part should be better written. I understand the idea of "predicting" the zero point offset using the two neighbour ones. What is not clear is: is this analysis used to remove "bad" zeroing point? If so, it should be expressed more clearly. Where the 2.1 ppb comes from? From my understanding zero point offsets will be temporally correlated if the difference between the predicted and the measured zero point value (which I will not call standard deviation) will fit within the 4.3 ppb expected in case of white noise. The quantiles are highly reported in this work but not enough used in the discussion.

- 25 **R** We computed:
  - Zero point offset at time i: ZPO (ti)
  - Difference of two consecutive ZPO: dZPO

If the ZPOs were normally distributed, their differences would be normally distributed as well and the standard deviation of the differences would be sqrt(2)\*sd(ZPO)=4.3 ppb. However, we found a standard deviation of only 2.1 ppb for the differences showing that consecutive ZPOs are temporally correlated. Temporal correlation is a requirement for the linear interpolation between the ZPOs.

The quantiles are reported in addition to the histograms in order to support the reader in comprehending the distributions shown in the figures.

35 **M** We suggest to keep the text as is.

**C** Page 11, Line 8: Why authors compare measurement from the tram and fix site with different time averages? For fig 6 is clear that measurements with 3 sec integration can show much high

variability (due probably to local production). To be more consistent, the comparison should be for the same time window and same averaging time. Authors should discuss about the higher variability observed at shorter time averaging (3 sec).

R In fact, figure 6 shows 5min averaged NO2 concentrations of the sites STA (red line), SCH (blue line) and 5min averaged concentrations measured by the tram instrument (black line). In this way the data can be well compared. Additionally, the data of the tram instrument is shown in full temporal resolution (grey). But the message of this plot is that the mobile instrument that regularly passes both fixed sites is in good agreement with the fixed sites when the same averaging is applied. We suggest keeping the text as it is.

10

**C** Page 11, Line 14: The statement "The calibration of sensors using precise measurements from AQM sites is an option for mobile operated sensors" requires better explanation.

**R** We added explanation to the respective section in response to *comment h*)3) of Reviewer #2 and now it should be clear.

15 **M** See reply to comment of Reviewer # 2.

**C** Table 1 Caption: replace number of values with number of measurements. Here too, the quantile values are not well used in the discussion. Authors should either used the more critically or remove them from the table.

20 **R** Thanks, caption has been changed. We suggest keeping the quantiles in the table, as they provide insights in the variation of the differences of mobile tram based measurements and the fixed air quality monitoring sites.

**M** Caption has been changed.

- **C** Page 11, Line 21: Suggestion to replace "aggregated by 50 meters of track length" with "with 50 meters resolution". Same in the Caption of Figure 8. As mentioned in the caption, the fact that in the figure are the results of the full campaign (and not only 1-way from Bellevue and Bahnofquai should be also mention in the text. Authors should give an idea if how many 1-way trips are used for this plot and how much this corresponds in term of time.
- R We suggest to keep only the key messages in the text and leave the more technical information about the number of underlying measurements (148 runs) in the figure caption.
   M Text has been changed as suggested.

C Figure 8. "The tram is directly impacted by traffic between the first and the third stop as well as
 between the fifth and the seventh stop. There is only little traffic between the third and the fifth stop." Not important in the caption, but this discussion should come out in the text, mentioning that there is then a correlation between the zones which are highly impacted by traffic and the

NO2 level measured. What are the "extra" dots in the graph? Out-layer data? This should be specify.

**R** We agree. Text and figure caption will be changed as suggested.**M** Text and caption of Figure 8 has been changed.

5

10

15

20

25

35

**C** Page 11, Line 21: what authors mean for similar locations? This is something that it should be explained more clearly. What are the criteria for making two locations similar?

**R** We reworded the sentence to improve understanding.

**M** The text has been changed to: "Our statistical models are driven by measurements and are based on the assumption of comparable  $NO_2$  concentrations at locations that are similar in terms of vicinity to sources (in particular traffic), the built environment and elevation."

**C** Figure 9. There is a miss of general explanation about this analysis. I have the feeling that all for plots at figure 9 don't bring a lot of information from the statistical point of view. I think authors should do a stronger effort in reporting data which are important for the understanding of the work. In the text this figure is not well descripted, ant the readers don't really get the point of what is plotted. Few more details: The really last red dot at the top right is suspicious... For this meas-urement period there should be a black dot (MAXTi) on top of the Q095Ti dot. Modelling period is a bit confusing because to me this is a simple data analysis, and there is not modelling behind. I suggest to change it with measurement period. The use of "Frequency" on the y-axes to me is also ambiguous, I would suggest to put "number of occurrences". I found weird that in figure c and d the y-axes does not extend to include the whole data set.

**R** In this case, we do not agree with Reviewer #3 and think that Figure 9 provides useful information about the spatio-temporal variability of  $NO_2$  measurements in the city of Zurich, as obtained from the mobile measurements. It is true, Figure 9 is somewhat dense and complicated but we feel that it is appropriately presented and explained in the text. We therefore suggest to leave Figure 9 and the corresponding discussion in the text as it is.

M No changes.

30 **C** Page 16, Line 9: "The frequency a particular variable is effectively used for splitting the 30 minutes data set depends only on the measurements". I guess authors wanted to say that "The frequency of a particular variable is effectively used for splitting the 30 minutes data set depending only on the measurements". Anyway this sentence should be rewritten more clearly.

**R** The reviewer is right. We could rephrase it as suggested but think that the sentence is not needed at all. It does not provide additional information for the reader.

**M** Sentence has been deleted without replacement.

**C** Page 16, Line 11: The whole paragraph: "The number and spatial distribution of measurements in a modelling period (approximately 500)... ...Regression trees based on tram measurements have about 6 location types Ri on average." should entirely be rewritten. I don't see how the number of measurements in the modelling period will limit the possibility of separate different location types?

- 5 **R** The concentration predicted by the statistical models is the mean of all the measurements referring to this location type. The higher the variability of NO2 concentration at a particular location the more measurements are required to accurately estimate the mean value. A representative set of measurements has to be assigned to each location type.
- M "The number of location types for which representative NO2 concentrations can be determined
   based on our statistical models is limited. The number of measurements required for estimating an accurate mean concentration for a location type depends on the short-term temporal variability of the NO2 concentration (see Figures 9 (c) and (d) and section 3.3.2). However, the number and spatial distribution of the measurements in a modelling period (approximately 500) is given by the tram route and cannot be influenced. Accordingly, the use of too many explanatory variables in our statistical models may lead to an over-determined model. Regression trees based on tram measurements have about 6 location types Ri on average."

**C** Section 3.3.1: I am sceptic on meaning of the model version Vt, especially on a such a short time scale. I guess that this approach could be les redundant by increasing the time intervals. For instance using the 30 min intervals to predict other 30 min intervals. Add "respectively" at the end of sentence (line 23).

**R** The statistical models rely on measurements and do not include any modeling features based on dispersion or chemistry. We selected the modeling time period to be 30 minutes in order to minimize effects from changing emission sources and meteorology. As models Vt perfom best in this modeling approach we concluded that there might indeed be potential for denser air quality measurement networks. We think that the conclusions are adequate to the performed analysis. We think that the comparison of the model is correct and adequately described and suggest to keep the text as it is.

**M** Line 23: The word "respectively" has been added as suggested.

30

20

25

**C** Page 16, Line 28: "The difference in performance between VAQ and VSTA is small due to the similarity of NO2 concentration encountered at site STA and by the tram-based instrument." Why? VAQ and VSTA are both based on fixed site. How come their similarity should be due to the similarity between NO2 measurement at STA and on the tram?

35 **R** While VSTA is based on data of AQM site STA only, VAQ uses data from all AQM sites including data from AQM site STA. Environment along tram tracks is most similar to that of STA in terms of traffic impact. That is why NO2 concentrations of site STA are taken in the modelling as predictions for the NO2 concentration for most tram positions. This explains why versions VAQ and VSTA only slightly differ.

**C** Table2: How the variables to be used on each model have been chosen? Why there are not variables for VSTA? Same question for section 3.3.2. From what authors describe it seems that variables are chosen to achieve the best overall fit to the measurements at AQM sites. But those variables have a specific meaning and they should not be selected to making the measurement better

matching but more according to their physical meaning.

5

10

15

20

25

30

**R** VSTA is based on data from site STA only (one site or location). Partitioning of the data in terms of geographic information is not possible.

We think that the variables used in the models (traffic intensities, SVF and elevation) have a relation to the instantaneous NO2 concentration field in Zurich. For example, the dependence of NO2 concentration on distance to a road depends on meteorology (beside other factors such as the built environment). Therefore, traffic intensities with different decay constants are included in the models. The idea of statistical modelling is to directly relate measurements with information that characterize locations. By validation methods such as leave-one-out the accuracy and limits of the derived models are tested.

**C** Figure 11 and its discussion: In my opinion, Figure 11 shows that model prediction is less effective at stations far away from the tram track (BLU and HEU). As expected, predictions that use data from the stations (WAQ and WT+A) provide better RMSE but still slope values are more far from unit. I don't think difference between models can be really discussed here. What about error bars in figure 11?

**R** The reviewer is right that the predictions based the statistical models are less accurate if a location largely differs from the locations of the measurements the model is based on in terms of traffic impact, building density or elevation. That is true for models based on mobile measurements as well as for models based on fixed AQM sites.

We compare in Figure 11 models based on the tram measurements, measurements from AQM sites as well as their combination. The fact that the model versions VAQ and VT provide similar results is important as VT is based on only one mobile instrument. Figure 11 further shows that if the spatial NO2 concentration is derived by a combination of measurements and statistical modeling the set of measurements must provide a good coverage of all the pollution situations. The AQM sites operated in the city of Zurich provide a good picture of the pollutant situation in Zurich although the differentiation of pollutant types is limited by the number of stations.

**M** We suggest to keep the text as it is.

35 **C** Section 3.4: "Our results suggest that the number of required mobile or static QCLAS instruments for comprehensive NO2 measurement in Zurich is not smaller than the current number of fixed AQM sites." How authors get to this conclusion?

**R** Section was probably unclear. We tried to make this point clearer.

**M** Following text has been added: "Due to the number of required measurements for location types with highly variable  $NO_2$  concentration, the necessity for measurements at locations with different pollutant situations and the travel time from one to another location, we suggest that the number of required mobile or static QCLAS instruments for comprehensive  $NO_2$  measurement in Zurich is not smaller than the current number of fixed AQM sites."

**C** Section 3.4: What about taxis? I guess those could provide a more random and spread-out data set with respect to public transportation which will follow a fix path.

R Yes, passenger cars could also be equipped with QCLAS sensors. However, a Tram or Bus has the advantage that an instrument can be installed permanently and sufficient power is available.
 M No changes.

**C** Page 19, Line 19: "The model type the maps are based on is irrelevant in this context." Meaning-less.

15 R We want to emphasize that the use of such an instrument for validation is independent from statistical models. However, we realized that this sentence can be misleading and is not important here. We therefore suggest to delete this sentence without replacement.

**M** Sentence has been deleted without replacement.

20 **C** Page 19, Line 23: "within the same footprint." What does it mean? Same source?

**R** We meant that with such lasers we could build an instrument of the same size that measures NO and NO2 simultaneously. We will change the wording.

**M** Text has been changed to "With the latest dual-wavelength QCL technology (Jagerska et al., 2014; Süess et al., 2016), the next generation of the QCLAS instrument could measure NO and NO2, simultaneously, with the same compact geometry."

**C** Technical comments:**

5

10

25

Page 2, Line 10: « Similarly, electrochemical sensors are currently not suited for such applications as their response time is not short enough ». Add a reference

- Page 2, Line 25: « a cylindrical multipass cell ». Put the reference Mangold 2016 also here.
  Page 2, Line 27: replace 1.1 with 2.1 Page 2, Line 30: remove Additionally
  Page 3, Line 14: change "it is not sensitive" with "it is less sensitive".
  Page 3, Line 17: « while the air flux is limited by a 100 μm diameter orifice ». Specify to which flow.
  Page 3, Line 22: « nano PC ». Specify which one.
  Page 3, Line 34: Say what GSM stays for.
  - Page 4, Line 8: « altering intervals ». I suggest to replace altering intervals with varying intervals. Page 6, Line 5: replace "in the environment of this location" with "in a specific location".

Page 6, Line 17: replace « for every three seconds » with "every 3 seconds ".

**R** We thank the reviewer for the technical comments and made the necessary changes.

**Spectroscopic real-time monitoring of NO2 for city scale modelling**

P. Morten Hundt1, Michael Müller1, Markus Mangold1,3, Béla Tuzson1, Philipp Scheidegger1, Herbert Looser1,2, Christoph Hüglin1 and Lukas Emmenegger1

[revised manuscript text omitted]

---

## Referee Report (RR1)

Comments on the manuscript AMT-2017-30

The manuscript "Spectroscopic real-time monitoring of NO2 for city scale modelling" by Hundt et al. describes the set-up of a new NO2 instrument and its use in the city of Zürich onboard a tram. The observational data gathered during 6 month of operation are compared to the results of nearby Air Quality Monitoring stations showing good overall performance of the instrument. Additionally, the data set is used in a statistical analysis together with spatial information of traffic intensity. Although the data set itself and the general approach seems to be promising I cannot recommend the current paper for publication due to logical deficiencies in the model analysis.

It is my opinion that the paper would be better-off if only the instrument, the general features of the observational data set (with DTV050 etc., see below), and the comparison to the AQM station data would have been presented. This in itself would be valuable information. The paper contains on the one hand a lot of detailed information, on the other hand significant information for the basic understanding of relationships is missing. Due to this imbalance the text of the paper is not simple to read and it is hard to stay tuned to gain conclusions from it.

The major gap in the statistical model analysis is the misunderstanding between the connection between NO2 concentration and traffic intensity. Car traffic primarily emits NO with a small addition óf NO2 depending on the type of motor (petrol or diesel). The NO2 contribution to the NOx emission of a car is on the order of 10-25%. At low traffic intensity, the NO from the car exhaust is oxidized to NO2 from ambient O3. This lays the ground for an almost linear relation between NO2 and traffic intensity at low NOx concentrations. In a main street, the traffic intensity is so large that all of the ambient O3 is titrated away by the NO emissions of the cars. In that region, the NO2 concentration is almost saturated at the level of the undisturbed ambient O3 concentration. This titration effect can clearly be seen in figure 6. As a consequence, the link between NO2 and traffic intensity is strongly non-linear at large NOx concentrations. In these cases, O3 goes to zero, NO2 goes to the concentration of O3 expected without traffic emissions, and NO can reach large concentrations, sometimes 10 times larger than NO2. As a consequence, a statistical analysis based on a linear relation between traffic intensity and NO2 will fail and will give apparent parameters very much depending on the range of NO2 concentrations included in the analysis. This effect is demonstrated in figure 10 of the current paper which shows the results of the model approach. In panel a, low and high traffic stations from the AQM network are used in combination with traffic intensity data. The streets are clearly visible in this map, but also nearby regions of the city. In panel b, the result of the tram observations following the relative large traffic in main-streets is used. As a consequence of the above mentioned NO2 saturation effect, the contrast between streets and residential areas is very large and the calculated NO2 in the streets totally dominates this map. In panel c, when the AQM (containing small and large NO2 regions) and the tram data-sets are combined, something between panel a and panel b is calculated. Which one is better? There is no measure to decide. If one would want to follow such an approach, the NOx concentration would be needed. And this is not possible to do with the current NO2-instrument used in the tram.

The mentioned imbalance between detailed information presented and information which is missing can be shown when looking at the instrument performance. Figure 1 to 7 show a lot of detailed information of the instrument and its comparison to the data of the AQM station. I am convinced that the instrument gives reliable data for NO2. The basic information used in the model analysis like

DTV050, DTM, SVF is not presented at all. Figure 8 (and 6) would have been an appropriate place to show the average traffic intensity and sky view factor alongside the track of the tram. Especially since figure 8 shows a very important result: the average NO2 concentration along the track of the tram is almost constant with a small variability of +-10%. This is somehow unexpected. One would expect larger variability due to traffic density patterns. And where is the imprint of the sky view factor meaning where is the imprint of building density? This is the kind of missing information from which the reader really would benefit.

Minor comments:

Figure 3: The colors for the data series Zero-Air and Dark-Noise seem to be interchanged in the legend inside the figure. If I read the text above it correctly, the Dark-noise should be linear while the Zero-air levels off.

Figure 2 supplement: I do not see any difference to figure 6 of the main paper. Both figures contain dotted lines for the O3 concentrations at two AQM stations.

In page 19, several figure numbers are mission in the text.

In page 19, the difference between model scenarios V and W is not clear. What is scenario W_B?

---

## Referee Report (RR2)

Editor's comments on the paper amt-2017-30
"Spectroscopic real-time monitoring of $NO_2$ for city scale modelling" by Hundt et al.

**General comments**

The paper presents a newly developed instrument (QCLAS) for the direct measurement of $NO_2$ using IR laser absorption spectroscopy. The technique is applied for mobile measurements on a tram in the city of Zurich in order to study the temporal and spatial distribution of $NO_2$ along selected roads. For quality control, measurements are compared with data from CLD instruments that are operated at fixed air-quality monitoring sites (AQMS). The measured data are then used as input for statistical models to predict spatial $NO_2$ concentration maps for the whole area of Zurich.

Major parts of the paper (the measurement technique, the instrumental comparisons and the model approach) are not satifactorily described and discussed. Furthermore, the presented material does not sufficiently support results and conclusions. In the abstract, the authors state: "Thorough analysis of the obtained $NO_2$ data, for instance by comparison with data from fixed air quality monitoring (AQM) sites, revealed the instrument to be highly accurate ...". First of all, I am missing a presentation of an independent characterization of the QCLAS instrument, as well as a thorough discussion of its measurement errors (see detailed comments). Second, it is pointed out that QCLAS is a direct technique which relies on Beer's Lambert law, whereas $NO_2$ measurements by CLD (used at the monitoring stations) are indirect, need calibration and can have a positive bias from interferences especially in polluted urban air. My expectation would be that QCLAS serves as an absolute reference against which the CLDs can be tested. Contrary, the authors remove a systematic 20 % discrepancy observed between QCLAS and CLD data by scaling the QCLAS measurements to the CLD data. Not surprisingly, thereafter the data sets agree very well (Figure 4). This approach does not support the general claim of a 'high accuracy' of the QCLAS technique presented in this paper. It remains unexplained why the authors have more trust in the CLD data rather than in their new absorption spectrometer. They assume inlet losses in the QCLAS, but what about errors and possible interferences in the CLD? Another very important point that remains unexplained is the reported offset drift of the $NO_2$ data measured by the QCLAS instrument. How can a concentration measurement based on Beer's Lambert law exhibit an offset drift?

The second part of the paper describes how $NO_2$ measurements from the tram and from monitoring stations are used to predict $NO_2$ concentration maps by statistical models. Here, I am missing a more detailed introduction of the concept for those readers who are not familiar with statistical models and regression trees. Although I understand that the approach does not need a full understanding of all processes contributing to the distribution of $NO_2$, some reasonable parameters have to be chosen in the model which are expected to correlate with $NO_2$. It is intuitive to chose parameters (e.g. DTV030, DTV100) related to traffic intensity, since traffic emissions are the dominant source of NOx in the city. However, what is the rationale for the "sky view factor" (SVF) and "elevation" (DTM)? An explanation or justification for the chosen input parameters (DTVnn, SVF, DTM) should be given. With respect to the model results, I am missing a critical discussion of the possible role of ambient ozone and solar radiation, which play a role for the concentration of atmospheric $NO_2$ during daytime. As pointed out by Referee #1, most NOx is emitted as NO and the resulting $NO_2$ concentration is dependent on ambient ozone. Furthermore, the partitioning of NOx into NO and $NO_2$ is influenced by solar UV radiation (here: $NO_2$ photolysis frequency). The photostationary state of $NO_2$ can change on timescales of minutes, for example, with cloudiness or if an air mass passes

from a bright place into a dark street canyon or vice versa. How would such changes on a short time scale influence the model predictions of $NO_2$?

The paper in its current version is not suitable for publication in AMT and requires major revisions. The two main topics, the description of the measurement technique and the application for statistical modelling, are both not adequately presented and discussed. I recommend to extend the description of the measurement technique according to my suggestions below. I recommend to remove the part about statistical modelling. If the authors want to keep the statistical modelling part, major revisions would be needed here too which include a discussion of the influence of $O_3$ and solar radiation on $NO_x$ and their relevance for the determination of maps of $NO_2$ concentrations in a city. Adequate presentation of both, the instrumental description and the statistical modelling, would certainly lead to a relatively long paper. Therefore, I recommend to leave out the model part here and consider to publish it as a revised version separately, for example, in ACP or a specialized journal on air pollution.

**Detailed comments**

*Introduction*

The overview of measurement techniques that are being used to measure atmospheric $NO_2$ is not complete. DOAS, LOPAP, LIF, luminol chemiluminescence should be mentioned. See for example, Villena et al., Atmos. Meas. Tech., 4, 1663–1676, 2011; Sluis et al., Atmos. Meas. Tech., 3, 1753–1762, 2010; Dari-Salisburgo et al., Atmos. Env. 43, 970–977, 2009; Dunlea et al., ACP 7, 2691, 2007; Thornton et al. Anal. Chem. 72, 528-539, 2000.

For the measurement of $NO_2$, the widely used CLD technique can have interferences in polluted air. This is not only the case for instruments with catalytic converters as mentioned in the introduction, but also for instruments with photolytic converters which can show interferences from PAN, HONO, VOCs (Reed et al., Atmos. Chem. Phys. 16, 4707, 2016 and Villena et al., AMT 5, 149, 2012).

*Instrument*

Some more technical details describing the instrument should be given, e.g. supplier and model of components (e.g. breadbord, etalon, mirrors, mirror holders, reflective coating of absorption cell), reflectivities of absorptioncell walls and of turning mirrors, laser beam diameter, spectral bandwidth of laser, total optical transmission of absorption cell.

*Evaluation of measured spectra*

The $NO_2$ concentrations are calculated according to Beer's Lambert law and converted into mixing ratios. The values of all experimental parameters needed for the calculation should be specified together with their errors and uncertainties (absorption path lengths, absorption cross section, pressure and temperature in the absorption cell). How large is the accuracy of the resulting $NO_2$ mixing ratios?

Figure 1c displays a measured absorption spectrum with $NO_2$ and $H_2O$ lines. Was the spectrum measured in synthetic air or in ambient air? Besides the intense absorption lines, there seem to be modulations like ripples in the spectrum. Are these caused by the instrument or by other weak absorbers? A spectrum of pure synthetic air should be shown for comparison.

How is the spectral baseline intensity ($I_0$, without $NO_2$ absorber) at the central position of the $NO_2$ line determined? What is the smallest detectable absorbance of the instrument and what is the corresponding limit-of-detection for $NO_2$?

p. 6, line 4-5: "Moreover, data were omitted when the $NO_2$ spectrum was not clearly identified by the processing algorithm." Which criteria are used to decide, if a spectrum can be used or not? Normally, if the absorption of the analyte is smaller than the noise of the baseline, fitted spectra should yield concentrations scattering around zero within their fitting error.

p. 9, line 12-15: "Based on these results, the measurement strategy was designed such that it applies a drift correction to the concentration measurements by measuring $NO_2$-free air for 2 min after 18 min of measuring. This offset-correction is achieved by subtracting the linear interpolation of the mean of the zero-air measurements before and after each 18 min measurement period." What is the cause for the offset drift and how large (ppbv?) were the corrections? How can a concentration measurement based on Beer's Lambert law have a drifting offset? Are there spectral interferences from other variable atmospheric compounds?

The processing of the measured atmospheric spectra, the determination of the $NO_2$ absorption cross section and the determination of the final $NO_2$ concentrations should be explained in detail. The QCLAS instrument should, in principle, allow absolute, calibration free measurements of $NO_2$. Have the authors tested the instrument against an independent $NO_2$ standard (e.g. certified gas mixture)? How was the agreement?

The performance data of the new instrument should be mentioned in the abstract: upper and lower limit-of-detection, time resolution, precision and accuracy. In the text, a table with specifications and instrumental parameters would be useful.

Could the instrument be used for absolute water vapor measurements ? Figure 1 shows an isolated $H_2O$ absorption line that could probably be used?

*Interferences*

Possible limitations and interferences should be discussed in more detail.

- o   Are there any known spectral interferences?
- o   The production of $NO_2$ from the reaction of NO and O3 in the inlet line and absorption cell is mentioned. Can you quantify the expected percentage range of perturbation for low $O_3$ (10 ppbv) and high $O_3$ (100 ppbv) conditions?
- o   The absorption cell is operated at reduced pressure favouring the thermal decomposition of labile components like PAN. Could this effect play a role in the instrument on a hot summer day in a polluted urban environment?
- o   Page 10: "... a correction to the normally calibration free QCLAS method was necessary because of loss of $NO_2$ during the sampling and in the long path cell... The contact of $NO_2$ with metal or plastic surfaces leads to dissociation." Is there direct, independent experimental evidence for such a wall loss in the instrument, or is this just an assumption? Which kind of materials were used? How long was the inlet line? Has the loss been checked by using different lengths of inlet lines or by variation of the volume flow rate of the sampled air? If the $NO_2$ wall loss is high, it could change over time with surface conditions (adsorbed $H_2O$, trace gases, particles). Are there any indications for such a behaviour?

*Measurement comparisons*

On page 10, the $NO_2$ measurements obtained by the QCLAS instrument and a CLD instrument are directly compared side-by-side at one AQMS station. The comparison and the conclusions drawn from it must be examined more critically.

The authors have apparently more confidence in the CLD measurements than in the QCLAS measurements and correct (calibrate) the QCLAS data to match the CLD measurements. This is somewhat surprising because the QCLAS is in principle an absolute technique, while the CLD needs calibration. Since the QCLAS measurements were scaled to the CLD data, Figure 4 cannot be said to demonstrate "excellent agreement". Rather, the scatter plot demonstrates a high linear correlation of the two techniques and a negligible offset. I suggest to plot the uncorrected QCLAS data against the CLD measurements and discuss linearity, slope and offset. The deviation of the slope (1.2) from unity should be discussed more critically with respect to known measurement errors of the QCLAS (see comments above) and CLD instruments (see comments below).  Only if these cannot explain the 20% difference, it is meaningful to speculate about other possible reasons.

The CLD technique is known to have several sources of measurement errors which are not discussed in the paper, but could contribute to the observed discrepancy. How was the CLD calibrated (against which standard?) and what was the accuracy of the calibration? The detection sensitivity of CLD instruments have a water vapor dependence. Was this corrected? CLD measurements of $NO_2$ rely on chemical conversion of $NO_2$ to $NO$. For catalytical and photolytic converters, interferences have been reported which can be quite significant in polluted urban air. To my knowledge, the Horiba APNA-370 instruments are equipped with catalytic converters which have a significant sensitivity to other nitrogen species (e.g. PAN, HONO etc.) yielding too high NOx readings. Have these interferences been characterized for the CLD used in this study and could these interferences explain the difference of 20 % between the QCLAS and CLD measurements?

Possible $NO_2$ losses in the inlet line of the QCLAS instrument are mentioned (s. above). What about possible losses in the inlet line of the CLD? Were other materials used than for the QCLAS instrument?

How far apart were the inlets of the two instruments? Were both inlets exposed to the same amount of direct and diffuse solar UV? If the $jNO_2$ values are different, the $NO_2/NO$ ratio may be different even if $NO_x$ is the same.

*Statistical model*

In addition to the report of Referee #1, I have the following question.

Has the "statistical model approach" ever been tested with simulated $NO_2$ concentrations from a regional atmospheric-chemistry model? One could simulate stationary or mobile measurements taking modelled $NO_2$ values or at fixed locations or along selected trajectories (roads), respectively, and superimpose artificially measurement noise. The statistical model could then be fed with simulated measurements and its output could be compared with the complete spatial $NO_2$ distributions from the atmospheric-chemistry model. The whole test could be done  for variable ozone background and varying solar radiation.

*Technical comments*

- p. 2, line 14: give reference for electrochemical sensors.
- p.7, definitions of DTV: do you mean "at a distance of..." or "within a distance of..."?
- p. 9, Figure 3: the explanations of the colors in the legend and in the figure caption disagree.
- Figure 4: are the errors of the fitted parameters 1sigma values?
- Figure 7: errors of the fit parameters should be given.
- p. 13, line 11: one of the reasons for the large scatter in Figure 7b is explained to be due to "a larger distance" than in Figure 7a. However, for both panels, a radius of 30 m is specified.
- NABEL 2016: where is the report available? The source, publisher, authors, ISSN etc. should be given in the reference.

---

## Editor Decision (ED1)

Editor's comments on the revised paper amt-2017-30 (v4)
"Mid-IR spectrometer for mobile real-time urban $NO_2$ measurements" by Hundt et al.

**Comments**

The paper has greatly improved. I have only a few minor comments which the authors should address before the paper can be published in AMT.

(1) My previous questions about the instrument performance are now answered in the revised paper, but the information is scattered in different chapters. A comprehensive summary should be given in the abstract and conclusion sections. In my understanding, the applied measurement technique has in principle a high potential for accurate, precise, and absolute measurements of $NO_2$. The $1\sigma$ precision is 0.13 ppbv at 3 s time resolution. The maximum uncertainty for the determination of $NO_2$ concentrations from Lambert-Beer's law is 5%. This accuracy can be expected under ideal conditions when the ambient temperature is constant and if NO2 wall loss can be excluded in the instrument. In practice, a temperature-dependent drift of optical interference fringes in the measured spectra was found to cause an $NO_2$ offset of up to +/- 5 ppbv, which was corrected based on regular baseline measurements with $NO_2$ free air. Furthermore, an $NO_2$ loss of 10% inside the instrument was quantified in laboratory experiments. For data consistency in the field study, the QCLAS data were adjusted by +20% to match the measurements of a reference instrument which belongs to the Zurich AQM network.

(2) Figure 6: are the fitting errors given as $1\sigma$ or $2\sigma$ values?

(3) Figure 7: I suggest to show the ozone measurements in the main paper instead of in the supplement. The ozone data are quite instructive as they indicate a nearly complete depletion of ozone when $NO_2$ reaches a maximum due to traffic emission of NO. In Figure 12, the local emission signal of $NO_2$ seems to become saturated at high traffic intensity above 5E+06. Could this be due to depletion of ozone such that emitted NO is not further contributing to $NO_2$? A comment about this behaviour should be given in the paper.

(4) Figure 12: in order to be consistent with Figure 11, the x-axis should be labelled as Traffic Intensity.

---

## Author Response (AR2)

Response to editor's comments:

We thank the editor for his detailed review of our revised manuscript. We fully agree with the suggestions and correspondingly we fundamentally re-structured and rewrote the paper. The major modification includes the removal of the statistical modelling part and extension of the technical description. Now, the paper is focused on the analytical instrument, its characterization, validation, and application. In the following, we address the specific comments individually.

**Detailed comments**

Introduction

The overview of measurement techniques that are being used to measure atmospheric NO2 is not complete. DOAS, LOPAP, LIF, luminol chemiluminescence should be mentioned. See for example, Villena et al., Atmos. Meas. Tech., 4, 1663–1676, 2011; Sluis et al., Atmos. Meas. Tech., 3, 1753–1762, 2010; Dari-Salisburgo et al., Atmos. Env. 43, 970–977, 2009; Dunlea et al., ACP 7, 2691, 2007; Thornton et al. Anal. Chem. 72, 528-539, 2000.

For the measurement of NO2, the widely used CLD technique can have interferences in polluted air. This is not only the case for instruments with catalytic converters as mentioned in the introduction, but also for instruments with photolytic converters which can show interferences from PAN, HONO, VOCs (Reed et al., Atmos. Chem. Phys. 16, 4707, 2016 and Villena et al., AMT 5, 149, 2012).

We added a corresponding paragraph and discussed the cross sensitivities of CLD detection and alternative $NO_2$ detection methods.

Instrument

Some more technical details describing the instrument should be given, e.g. supplier and model of components (e.g. breadbord, etalon, mirrors, mirror holders, reflective coating of absorption cell), reflectivities of absorptioncell walls and of turning mirrors, laser beam diameter, spectral bandwidth of laser, total optical transmission of absorption cell.

The instrument description now features the requested details.

Evaluation of measured spectra

The NO2 concentrations are calculated according to Beer's Lambert law and converted into mixing ratios. The values of all experimental parameters needed for the calculation should be specified together with their errors and uncertainties (absorption path lengths, absorption cross section, pressure and temperature in the absorption cell). How large is the accuracy of the resulting NO2 mixing ratios?

An estimation of the total error of the spectroscopic measurement was added and discussed.

Figure 1c displays a measured absorption spectrum with NO2 and H2O lines. Was the spectrum measured in synthetic air or in ambient air? Besides the intense absorption lines, there seem to be modulations like ripples in the spectrum. Are these caused by the instrument or by other weak absorbers? A spectrum of pure synthetic air should be shown for comparison.

For clarity, we replaced the spectrum with two spectra recorded using NO2-free air and ambient air, respectively. Both spectra show the water absorption line at $1600.47 cm^{-1}$.

The periodic ripples in the spectrum originate from optical interference fringes occurring in the multi-pass cell and the associated optical elements.

How is the spectral baseline intensity ($I_0$, without NO2 absorber) at the central position of the NO2 line determined? What is the smallest detectable absorbance of the instrument and what is the corresponding limit-of-detection for NO2?

The fitting routine includes the determination of a baseline by fitting a polynomial (up to order 4) to the spectrum left and right of the absorption feature. The normalization of the transmission spectrum is based on associated zero-level (laser off) measurements.

The residual of the Voigt profile fit is $3\times10^{-4}$ at 1s averaging time. The 1σ detection limit can be retrieved from the Allan-Werle Deviation plot: At 1s it is 300 ppt and the minimum is at 30 ppt after 200 s averaging.

p. 6, line 4-5: "Moreover, data were omitted when the NO2 spectrum was not clearly identified by the processing algorithm." Which criteria are used to decide, if a spectrum can be used or not?

Normally, if the absorption of the analyte is smaller than the noise of the baseline, fitted spectra should yield concentrations scattering around zero within their fitting error.

The water absorption line shown in the spectrum was used to determine whether the QCL is operating at the right frequency. When the measured water concentration was not within the expected range the data have been discarded.

As the water line is always detectable in the spectrum, it was used as a frequency marker to allow fitting of the NO2 spectral region even if no detectable NO2 concentration was present. This results indeed in scattering around zero within the fitting error. The zero-point however may drift (caused by interference fringes) and therefore is regularly determined and later corrected for by measuring filtered NO2 free air.

There are no spectral interferences. To ensure data quality, data where the zero-point correction exceeds ±5 ppb were excluded from the analysis.

p. 9, line 12-15: "Based on these results, the measurement strategy was designed such that it applies a drift correction to the concentration measurements by measuring NO2-free air for 2 min after 18 min of measuring. This offset correction is achieved by subtracting the linear interpolation of the mean of the zero-air measurements before and after each 18 min measurement period." What is the cause for the offset drift and how large (ppbv?) were the corrections? How can a concentration measurement based on Beer's Lambert law have a drifting offset? Are there spectral interferences from other variable atmospheric compounds?

We rewrote the corresponding paragraph for more clarity. As mentioned above interference fringes originating from the optical setup cause a structure on the retrieved spectra that are interpreted as absorption and thus cause an offset from zero in NO2 free air.

The processing of the measured atmospheric spectra, the determination of the NO2 absorption cross section and the determination of the final NO2 concentrations should be explained in detail.

We extended the paragraph describing the determination of NO2 concentrations.

The QCLAS instrument should, in principle, allow absolute, calibration free measurements of NO2. Have the authors tested the instrument against an independent NO2 standard (e.g. certified gas mixture)?

How was the agreement?

Yes, the instrument should in principle be calibration free. However the uncertainties in cross section, path-length, pressure and temperature lead to a systematic error that makes a calibration necessary. Since calibration with NO2 reference gases is a tricky and currently not established issue. We therefore decided to use a CLD (despite its shortcomings and possible cross sensitivities) as a reference method. This calibration strategy allows a comparison of the mobile QCLAS measurements on top of the tram with the CLD measurements at the two air quality monitoring sites situated next to the tram tracks. We rewrote the corresponding paragraph.

The performance data of the new instrument should be mentioned in the abstract: upper and lower limit-of-detection, time resolution, precision and accuracy. In the text, a table with specifications and instrumental parameters would be useful.

We mention the instrument's precision, time resolution and dynamic range in the abstract.

Could the instrument be used for absolute water vapor measurements? Figure 1 shows an isolated H2O absorption line that could probably be used?

Yes, indeed. In fact, the water concentration retrieved by the fitting algorithm was used as a check to make sure that the laser operates in the right spectral region.

Possible limitations and interferences should be discussed in more detail.

o  Are there any known spectral interferences?

No.

o  The production of NO2 from the reaction of NO and O3 in the inlet line and absorption cell is mentioned. Can you quantify the expected percentage range of perturbation for low O3 (10 ppbv) and high O3 (100 ppbv) conditions?

The reaction of NO and O3 is fast and leads to formation of NO2 in the immediate vicinity of the source of NO emissions (i.e. along roads). In the inlet line of instruments, this reaction can continue and additional NO2 can be formed. We calculated the formation of NO2 in the inlet line of the reference CLD instrument at the urban traffic site SCH in Zurich, where concentrations of NO and O3 are available. The residence time of the air sample in the inlet line of the CLD is about 5 s. For the time period discussed in the paper (September 21, 2015 until March 9, 2016) we find an average NO2 formation in the inlet line of 0.5 ppb, the average NO2 concentration during that time period at SCH was 24.3 ppb. For the conditions at SCH, the NO2 formation in the inlet and absorption cell of the QCLAS would be somewhat larger, because of the longer residence time of the air sample (about 10 s), for the QCLAS we calculated an average NO2 formation of 1.0 ppb. Note that the formation of NO2 in the inlet line depends on the O3 concentration, but also on the concentration of NO. The maximum formation of NO2 in the inlet line does not occur at peak ozone conditions, because NO concentrations are then typically low. Largest NO2 formation in the inlet occurs at conditions with moderate but sufficient levels of both O3 and NO. The NO2 formation in inlet lines due to reaction of NO and O3 is largest at locations that are strongly impacted by traffic (or other sources of NO). The SCH site is an urban traffic site next to a major road and the calculated formation of NO2 can be considered as an upper limit for NO2 formation in the inlet system of the QCLAS during the described mobile deployment.

We suggest not going into details here, because NO2 formation in inlet lines is most of the time a small perturbation and occurs in the QCLAS but also in the reference CLD.

o The absorption cell is operated at reduced pressure favouring the thermal decomposition of labile components like PAN. Could this effect play a role in the instrument on a hot summer day in a polluted urban environment?

This effect could happen but would lead to a very minor interference. Measurements within the Swiss Air Quality Monitoring network show that PAN concentrations in the Zurich area reach during summer maximum hourly concentrations of 1.5 ppb, and concentrations well below 1 ppb during the cold season, when the mobile measurements with the QCLAS have been performed. Note that PAN is one of the interfering compounds when measuring NO2 with CLD instruments.

o Page 10: "... a correction to the normally calibration free QCLAS method was necessary because of loss of NO2 during the sampling and in the long path cell... The contact of NO2 with metal or plastic surfaces leads to dissociation." Is there direct, independent experimental evidence for such a wall loss in the instrument, or is this just an assumption? Which kind of materials were used? How long was the inlet line? Has the loss been checked by using different lengths of inlet lines or by variation of the volume flow rate of the sampled air? If the NO2 wall loss is high, it could change over time with surface conditions (adsorbed H2O, trace gases, particles). Are there any indications for such a behaviour?

The NO2 loss in the multipass cell has been observed experimentally, were the NO2 concentration of a gas flow through an inlet line was monitored by a CLD while passing or bypassing the multipass cell. A

systematic loss of 10% in the latter case has been repeatedly measured. However, no systematic survey of different flows has been done, except the average condition that were used for the field campaign. Nevertheless, similar effects were found for dry NO2 calibration gas.

Measurement comparisons

On page 10, the NO2 measurements obtained by the QCLAS instrument and a CLD instrument are directly compared side-by-side at one AQMS station. The comparison and the conclusions drawn from it must be examined more critically.

The authors have apparently more confidence in the CLD measurements than in the QCLAS measurements and correct (calibrate) the QCLAS data to match the CLD measurements. This is somewhat surprising because the QCLAS is in principle an absolute technique, while the CLD needs calibration. Since the QCLAS measurements were scaled to the CLD data, Figure 4 cannot be said to demonstrate "excellent agreement". Rather, the scatter plot demonstrates a high linear correlation of the two techniques and a negligible offset. I suggest to plot the uncorrected QCLAS data against the CLD measurements and discuss linearity, slope and offset. The deviation of the slope (1.2) from unity should be discussed more critically with respect to known measurement errors of the QCLAS (see comments above) and CLD instruments (see comments below). Only if these cannot explain the 20% difference, it is meaningful to speculate about other possible reasons.

No, we do not have more confidence in the CLD measurements. The point is that calibration of direct and specific NO2 instruments using NO2 reference gases is still an unsolved issue. There are reference gases available (either static ones in cylinders or dynamically generated NO2 using permeation sources), but it is difficult to avoid losses of NO2 due to wall effects and therefore to produce a reliable reference. As emphasized in the revised manuscript, we therefore decided to correct the QCLAS to match the CLD. Another advantage of this strategy is that the QCLAS measurements can be directly compared with the CLD measurements at the two AQM sites that were regularly passed by the tram (see revised manuscript).

As suggested by the reviewer, Fig. 6 shows now the relation between uncorrected QCLAS and CLD, the reasons for the observed differences between QCLAS and CLD are discussed in more detail.

The CLD technique is known to have several sources of measurement errors which are not discussed in the paper, but could contribute to the observed discrepancy. How was the CLD calibrated (against which standard?) and what was the accuracy of the calibration? The detection sensitivity of CLD instruments have a water vapor dependence. Was this corrected? CLD measurements of NO2 rely on chemical conversion of NO2 to NO. For catalytical and photolytic converters, interferences have been reported which can be quite significant in polluted urban air. To my knowledge, the Horiba APNA-370 instruments are equipped with catalytic converters which have a significant sensitivity to other nitrogen species (e.g. PAN, HONO etc.) yielding too high NOx readings. Have these interferences been characterized for the CLD used in this study and could these interferences explain the difference of 20 % between the QCLAS and CLD measurements?

In the revised manuscript we discuss the cross sensitivities of CLD instruments equipped with catalytic (molybdenum) converters in more detail. However, one should keep in mind that this measurement technique is the reference method in Europe and is mandatory for regulatory air quality measurements. The interferences of the CLD instrument have not been characterized within the scope of this study, but from earlier work we know that the conversion efficiency for other nitrogen compounds such as PAN and HONO is very high. From parallel measurements with two CLD instruments, one equipped with a catalytic and one equipped with a photolytic converter, we know that the interference of the standard CLD instruments cannot explain the observed difference between QCLAS and CLD. In Zurich (urban background) we find in an unpublished study, that on annual average 1.7 ppb (or 10%) of measured total NO2 can be attributed to interfering compounds. This is explained in the revised manuscript. CLD instruments are calibrated against NO reference gases. This is also mentioned in the revised manuscript. The contribution of the uncertainties of the calibration gases (primary and transfer standard) to the total uncertainty budget is small. On the other hand, the interference of CLDs to water as mentioned by the reviewer is the largest component of the total uncertainty of NO2 measurements using CLD (see NABEL 2016 for the detailed uncertainty budget estimation). The CLD measurements are not corrected for the interference of water, because the relationship is unknown.

Possible NO2 losses in the inlet line of the QCLAS instrument are mentioned (s. above). What about possible losses in the inlet line of the CLD? Were other materials used than for the QCLAS instrument?

Losses of NO2 in the inlet line of CLD instruments are not known. As outlined above, it is currently difficult to provide instruments with precisely known NO2 reference gases. For CLD measurements within regulatory AQ measurements it is currently only recommended to use materials for the sampling line that are fit for this purpose (e.g. PTFE), see e.g. EN 14211 (CEN, 2012).

How far apart were the inlets of the two instruments? Were both inlets exposed to the same amount of direct and diffuse solar UV? If the NO2 values are different, the NO2/NO ratio may be different even if NOx is the same.

The QCLAS was placed directly next to the inlet of the NABEL station (d<0.5m). Both were exposed to the same solar radiation.

Statistical model

In addition to the report of Referee #1, I have the following question.

Has the "statistical model approach" ever been tested with simulated NO2 concentrations from a regional atmospheric-chemistry model? One could simulate stationary or mobile measurements taking modelled NO2 values or at fixed locations or along selected trajectories (roads), respectively, and superimpose artificially measurement noise. The statistical model could then be fed with simulated measurements and its output could be compared with the complete spatial NO2 distributions from the atmospheric-chemistry model. The whole test could be done for variable ozone background and varying solar radiation.

Following the Reviewers' recommendation the modelling part has completely been removed from the publication.

Technical comments

- p. 2, line 14: give reference for electrochemical sensors.
- p.7, definitions of DTV: do you mean "at a distance of..." or "within a distance of..."?
- p. 9, Figure 3: the explanations of the colors in the legend and in the figure caption disagree.
- Figure 4: are the errors of the fitted parameters 1sigma values?
- Figure 7: errors of the fit parameters should be given.
- p. 13, line 11: one of the reasons for the large scatter in Figure 7b is explained to be due to "a larger distance" than in Figure 7a. However, for both panels, a radius of 30 m is specified.
- NABEL 2016: where is the report available? The source, publisher, authors, ISSN etc. should be given in the reference.

In the revised manuscript, all technical comments have been addressed.

---

## Author Response (AR3)

Response to editor's comments:

We thank the editor again for the critical review of our revised manuscript and for his commitment and dedication during the whole review process. We appreciate the way our submission was handled. In the following, we address the points that were raised.

(1) My previous questions about the instrument performance are now answered in the revised paper, but the information is scattered in different chapters. A comprehensive summary should be given in the abstract and conclusion sections. In my understanding, the applied measurement technique has in principle a high potential for accurate, precise, and absolute measurements of $NO_2$. The 1σ precision is 0.13 ppbv at 3 s time resolution. The maximum uncertainty for the determination of $NO_2$ concentrations from Lambert-Beer's law is 5%. This accuracy can be expected under ideal conditions when the ambient temperature is constant and if $NO_2$ wall loss can be excluded in the instrument. In practice, a temperature-dependent drift of optical interference fringes in the measured spectra was found to cause an $NO_2$ offset of up to +/- 5 ppbv, which was corrected based on regular baseline measurements with $NO_2$ free air. Furthermore, an $NO_2$ loss of 10% inside the instrument was quantified in laboratory experiments. For data consistency in the field study, the QCLAS data were adjusted by +20% to match the measurements of a reference instrument which belongs to the Zurich AQM network.
We modified the abstract and conclusion to give a comprehensive summary of the instrument's accuracy and the adjustment that we applied to the data.

(2) Figure 6: are the fitting errors given as 1σ or 2σ values?
The fitting errors are 1σ values. We added that information.

(3) Figure 7: I suggest to show the ozone measurements in the main paper instead of in the supplement. The ozone data are quite instructive as they indicate a nearly complete depletion of ozone when $NO_2$ reaches a maximum due to traffic emission of NO. In Figure 12, the local emission signal of $NO_2$ seems to become saturated at high traffic intensity above 5E+06. Could this be due to depletion of ozone such that emitted NO is not further contributing to $NO_2$? A comment about this behaviour should be given in the paper.
We added the ozone measurements of the AQM stations STA and SCH to Figure 7.
Figure 12 shows the local emission signal with respect to traffic intensity during time periods when the tram was operating. The intent of Figure 12 is to show that the measurements of our instrument are in line with measurements from AQM stations also with respect to traffic intensity.
The relation between traffic intensity and local concentrations of NO, $NO_2$ and $O_3$ is an interesting subject but the data from this study cannot substantially contribute to elucidate this topic. We think that for this purpose a temporally highly resolved (<10 minutes) data set including traffic, meteorological and pollutant data is required for several AQM sites. For example, AQM site SWD is located directly at a motorway that goes through a residential area. Pollutant concentrations at this site depend significantly on wind direction and wind speed. However, for producing Figure 12 no filters were applied to the measurements except from the time periods of tram operation.

Therefore, we prefer to refrain from discussing the relation between traffic and NO, $NO_2$ and $O_3$ concentrations.

(4) Figure 12: in order to be consistent with Figure 11, the x-axis should be labelled as Traffic Intensity.

Done.

[revised manuscript text omitted]